# Benchmark Dataset for Radiology Report Generation with Instructions and Contexts

## Abstract

While automatic report generation has demonstrated promising results using deep learning-based methods, deploying these algorithms in real-world scenarios remains challenging, where models may be required to follow the instruction from the radiologists and consider contextual information. Such instructional report generation tasks are critical for enabling more accurate, customizable, and scalable report generation processes, but remain under-explored and lack substantial datasets for training and evaluation. However, constructing a dataset for report generation with instructions and contexts is challenging due to the scarcity of medical data, privacy concerns and the absence of recorded user-model interactions. To tackle this challenge, we propose a unified and automatic data generation pipeline which leverages large language model (LLM) to produce high-quality instructions and context for report generation tasks. We present a new benchmark dataset *MIMIC-R3G* that extends the largest existing radiology report generation dataset MIMIC-CXR, comprising five representative tasks pertinent to real-world medical report generation. We conducted an extensive evaluation of state-of-the-art methods using the proposed benchmark datasets. Additionally, we introduced a baseline method, the Domain-enhanced Multimodal Model (*DeMMo*), demonstrating that leveraging training data containing instructions and contextual information significantly improves the performance of instructional report generation tasks.

## 1 Introduction

Radiology report generation is one of the straightforward yet essential task in computer-aided diagnosis (CAD) systems. It aims to automatically generate a text description of the patient's radiology images including professional medical diagnosis. Recent works can automatically generate radiology report accurately within seconds, which largely reduces the workload of professional radiologists in clinical routines (Jing et al., 2018; Chen et al., 2020; Liu et al., 2021; Wang et al., 2022a; Huang et al., 2023).

Most previous works treat radiology report generation as a captioning task, where a text decoder generate medical report based on extracted image features (Nicolson et al., 2023). In real clinical practice, however, the scenario and procedure might be more complex than a straightforward captioning task. Specifically, in real-world scenarios, the model is required to follow broader instructions of the radiologists and to consider different types of context information. For example, radiologists usually need to refer to the patient's X-ray images and reports from previous visits in order to write a more comprehensive report that includes progress or changes in the abnormalities. Also in many cases, patients are required to undergo some other medical examinations beside radiology screenings. All these kinds of extra information could affect how radiologists read the radiographs and write the final report for the patient. Therefore, this paper focuses on developing a practical report generation dataset that supports real-world clinical practice containing various interactions and context information.

To facilitate research on radiology report generation with instructions and context, a benchmark dataset needs to be developed that includes not only medical images and reports, but also rich contextual information and interaction data between doctors and report generation models. However, the scarcity of medical data, and the privacy concerns surrounding patient information in the public

domain, poses significant challenges. Also, current medical report generation datasets are predominantly obtained from hospital or clinical databases. The information available in these datasets is generally limited to medical images and associated structured reports (Johnson et al., 2019; Demner-Fushman et al., 2016), lacking supplementary contextual information that is essential for a thorough analysis and might influence radiologist's reasoning in formulating a diagnosis. Furthermore, collecting interaction data between doctors and report generation models is exceptionally costly, which requires integrating model deployment into clinical workflows without disrupting patient care, as well as extensive coordination with medical professionals.

To address the challenges, we examine the clinical requirements and propose an automatic data generation pipeline and a new benchmark dataset, named *MIMIC-R3G* (Real-world Radiology Report Generation). *MIMIC-R3G* contains five representative tasks pertinent to the medical report generation context: report generation with no context, report revision, template-based report generation, report generation based on patient's previous visits, and report generation incorporating patient's other information including medical records and laboratory tests. Building on these tasks, we introduce a unified automatic data generation pipeline to generate instructions, context, and reports in accordance with the ground truth report and images, using specific system messages and ground truth reports as input to direct large language model (OpenAI, 2022) for generation.

Furthermore, we introduce a baseline method, *DeMMo* (Domain-enhanced Multimodal Model), tailored for the proposed context-aware report generation tasks with various instruction inputs. This approach efficiently fine-tunes Flamingo model (Alayrac et al., 2022) by integrating a domain-specific medical vision encoder and incorporating additional pathological guidance. Comprehensive experiments on the *MIMIC-R3G* benchmark demonstrate that our method achieves promising results on all real-world report generation tasks, compared to state-of-the-art medical domain visual-language models.

In summary, the contributions of this paper are as follows:

- We present a new problem setting for real-world report generation that emulates clinical practices by incorporating various clinical interactions and contextual information.
- We propose the first real-world report generation benchmark dataset *MIMIC-R3G*, where a unified framework is designed to automatically generate the requisite context data, leveraging the power of LLM.
- We develop *DeMMo*, a large multimodal model with domain-specific capability enhanced via incorporating a general domain Flamingo with an additional medical vision encoder and pathological information for further guidance, serving as a baseline for the benchmark dataset.

## 2 RELATED WORKS

**Report Generation** Traditional methods use an encoder-decoder regime, where an encoder is used to extract image features, and a decoder is used to generate text from the features. The combination of CNN encoder and RNN decoder were utilized in earlier works (Jing et al., 2018; Xue et al., 2018; Wang et al., 2018; Hou et al., 2021). With the advent of Transformer architecture, researchers have explored the use of Transformer with specialized memory or attention mechanisms for report generation (Cornia et al., 2020; Chen et al., 2020; 2021; You et al., 2021). To further improve performance, many works incorporated pre-extracted pathology labels and domain-specific knowledge graphs as priors in the generation pipeline (Liu et al., 2021; Wang et al., 2022b; Huang et al., 2023; Li et al., 2023d). Some retrieval-based approaches have also gained prominence in recent years (Endo et al., 2021; Jeong et al., 2023). These methods predominantly employ contrastive learning techniques to retrieve probable texts from the training set as inference outcome. Building on existing approaches, several studies (Wu et al., 2022; Zhu et al., 2023) have also taken real-world clinical scenarios into account, but primarily focusing on the single task of incorporating reports from previous visits as a generation prior. We expand on this and propose a unified task formulation of real-world report generation.

**Large Language Models** With the strong ability in natural language processing and generation, Large Language Models (LLMs) have shown significant potentials in performing real-world report generation tasks. State-of-the-art LLMs (Brown et al., 2020; Touvron et al., 2023; Chowdhery

et al., 2022) are highly interactive and capable of following instructions for various language tasks (Ouyang et al., 2022), making it poses high potential in dealing with real-world clinical scenarios. Furthermore, the extensive volume of training data equips LLMs with the capacity to internalize domain-specific knowledge and exhibit reasoning capabilities within the medical field. Without fine-tuning on specific medical dataset, ChatGPT (OpenAI, 2022) is tested to pass the US Medical Licensing Exams (USMLE) (Kung et al., 2023) showing its promising ability to reason and process language in the medical domain. Finally, LLMs demonstrate proficiency in generating more extensive and complex text sequences, making them well-suited for medical report generation tasks.

## 3 Radiology Report Generation with Instructions and Contexts

In contrast to conventional report generation models, Real-world Radiology Report Generation (R3G) poses two significant differences. Firstly, it necessitates the model to adhere to the user's requests and instructions. Secondly, in addition to the medical image itself, the model must possess the capability to comprehend and utilize external contextual information in order to produce a more precise report. As a results, we propose several representative sub-tasks that resembles these two requirements, all of which are essential features widely applicable in clinical practice. The instances drawn from these representative sub-tasks will be used to train and evaluate our proposed report generation model.

**No Context Report Generation** This sub-task is the conventional report generation task without any additional instructions from radiologist or context information.

**Report Revision** Reports generated models may be sub-optimal in some cases, and and human professionals are still required to review and revise the output reports prior to submission. Therefore, it is desirable for the model to possess the capability of revising the report based on straightforward instructions to further alleviate the workload of the human professional.

**Template** In real-world scenarios, clinics or hospitals may employ structured report templates. These templates may comprise a list of common abnormalities or regions, and the radiologist is required to fill in the corresponding findings or absence of abnormalities. In sum, we want the model to be capable of generating report following any form of input template.

**Previous Radiology Image and Report as Context** In typical clinical practice, patients undergo multiple radiology screenings. It is essential for radiologists to write medical reports that not only focus on the current radiology image but also reference the patient's previous medical images and reports. This approach enables the production of a more informative report that can address the alterations in the disease progression compared to previous visits.

**Medical Records and Lab Tests as Context** Patient's medical records, including medical condition history, along with medical exams like blood tests and pulmonary function tests, are vital for accurate diagnosis. Medical records and lab tests are all crucial context information for radiologists to write reports, so the model should also posses the ability generate reports based on them.

## 4 *MIMIC-R3G*: Dataset for Report Generation with Instructions and Contexts

### 4.1 Task Formulation

We formulate the proposed real-world report generation tasks under a unified instruction-following paradigm, so we can fully utilize the instruction-following capabilities of a Large Language Model (LLM). Specifically, we format the proposed real-world report generation tasks into a unified single-round instruction-following example: $(V_i, I_i, C_i, R'_i)$, representing the $i$-th example in the dataset, where $V_i$ denotes a set of medical images; $I$ denotes the instruction from the user; $C_i$ refers to the context information provided to facilitate the report generation; and $R'_i$ refers to the ground truth report associated with the medical images $V_i$, instruction $I_i$, and context $C_i$ in the generated dataset. For all the sub-tasks, $V_i$ is directly utilized from the dataset.

### 4.2 Data Generation

Existing large-scale report generation datasets, such as MIMIC-CXR (Johnson et al., 2019), are not tailored for real-world report generation as they lack user instructions $I_i$ and contextual information $C_i$ paired with corresponding responsive report $R'_i$. The manual collection of such instructional

and contextual data is prohibitively costly and may raise privacy concerns. Hence, we propose to harness the capabilities of GPT and construct a unified pipeline to automatically generate diverse and relevant real-world clinical text data based on existing ground truth reports in conventional datasets.

The primary goal is to either design or generate instructions $I_i$ and context $C_i$, and also possibly modify the ground truth report $R_i$ from dataset into $R'_i$ according to different sub-tasks. To generate a dataset of a single real-world report generation task, the objection of our pipeline is $\{(V_i, R_i)\}_{i=1}^N \mapsto \{(V_i, I_i, C_i, R'_i)\}_{i=1}^N$, where $N$ is the number of examples of an existing report generation dataset.

The medical image $V_i$ stays un-changed and directly comes from the original dataset. We devise different task-specific system messages to generate the required $I_i$, $C_i$, and $R'_i$ for distinct tasks. Using the ground truth report $R_i$ as input, along with in-context examples (omitted in examples) to guide the output format, the response can be filtered and parsed accordingly into the required data components. For better accuracy, the generation, filtering, and auto-validation (explained later) are split into multiple rounds of GPT queries. Next we will elaborate on how request from each sub-task is organized as an instruction-following example, and how the examples are produced for each sub-task. We use OpenAI Chat Completions API with gpt-4-32k as the underlying engine in our generation pipeline. We show one example of data generation for report revision, and other examples, along with prompts for auto-validation are shown in the Appendix.

**Report Generation** For basic report generation task without context, the data sample follows $(V_i, I_i, C_i, R'_i)$, where $V_i$ and $R'_i = R_i$ are directly utilized from report generation dataset. $I_i$ is a manually designed instruction telling the model to generate the report based on given images, and $C_i$ is kept empty.

**Report revision** For report revision task, $R'_i = R_i$ come from the report generation dataset, $I_i$ is the instruction of how to revise or correct the report, and $C_i$ is the report that the user wants the model to revise. To generate $I_i$ and $C_i$ for this task, we employ our proposed pipeline to produce a slightly modified report based on the input ground truth report, along with the instructions of how to revise the modified report into the correct ground truth report.

**Template** $I_i$ is a manually designed instruction, *e.g.*, *Fill in the template based on the give medical images*. $C_i$ and $R'_i$ are the empty template and the corresponding filled template. We collect 10 report templates with help of medical professionals. 6 of them are from real-world sources, and 4 of them are generated using GPT-4. We leverage our pipeline to generate the structured version of the ground truth report based on the given templates. All templates are reviewed and revised by medical professionals, with details shown in the Appendix.

**Previous Visit as Context** $I_i$ is manually designed instruction telling the model to generate report based on both the medical images and report from last visit. $C_i$ can be the retrieved previous report of the same patient from the dataset, and $R'_i = R_i$ is the ground truth report. It should be noted that $V_i$ in this task can include medical images of the patient from their previous visit as well.

**Medical Records and Lab Tests as Context.** Similarly, $R'_i = R_i$ comes from the original dataset, and $I_i$ is a manually designed instructions. $C_i$ here represents the additional medical conditions or medical examination results that the patient may possess. Since the ground-truth medical report exhibits a strong correlation with the external context information, our pipeline generates $C_i$ by extracting medical indications and inferring the plausible medical conditions, medical examinations and exam results based on the ground truth medical report.

---

**Example of Generating Report Revision Data**

**System Message:** You are a radiologist and will intentionally make some diagnostic errors in a chest X-ray radiology report. You will be given a correct findings section of the chest X-ray report, and you will be asked to rewrite the report with a few, around 2-4, diagnostic errors. Provide the rewritten, incorrect report only.

**User (GT report):** PA and lateral views of the chest provided. The lungs are adequately aerated. There is a focal consolidation at the left lung base adjacent to the lateral hemidiaphragm. There is mild vascular engorgement. There is bilateral apical pleural thickening. The cardio mediastinal silhouette is remarkable for aortic arch calcifications. The heart is top normal in size.

**Response:** PA and lateral views of the chest provided. The lungs are adequately aerated. There is no focal consolidation. There is mild vascular engorgement. There is no bilateral apical pleural thickening. The cardio mediastinal silhouette is remarkable for aortic arch calcifications. The heart is mildly enlarged in size.

- - - - - - - - - - - - - - - - - - - - - - - - - - - - - - - - - - - - - - - - - - - - -

**System Message:** You are a radiologist who helps check the correctness of chest x-ray radiology report. Given a correct report and an incorrect report, provide 2 to 3 short and concise pointers on how to fix the incorrect report into the correct report. Do not be verbose or verbatim. List the instructions or suggestions only. The doctor has access to the radiograph, therefore your instructions could be vague or

only state what the problems are without actual instruction to fix them.
**User:** Incorrect report: ⟨previous response⟩ Correct report: ⟨GT report⟩
**Response:** 1. Add focal consolidation. 2. The patient has bilateral apical pleural thickening.

- - - - - - - - - - - - - - - - - - - - - - - - - - - - - - - - - - - - - - - - - - -

$I_i = $⟨Manually designed instruction and text in second response⟩
$C_i = $⟨Text in first response⟩
$R'_i = $⟨GT report⟩

**Quality Control of the Generation** Since the ground truth report $R'_i$ is either identical to original report $R_i$ or rewritten by GPT while preserving the medical diagnosis intact, our pipeline is able to produce accurate data with very few factual errors. Furthermore, the generated data has undergone both automatic and manual data quality exam and control processes. Specifically, after generating the data, GPT will be prompted again with the generated data as input and is required to check correctness. For report revision task, GPT checks the correctness of revision instructions. For template task, it checks whether the generated ground truth follow the diagnosis of original ground truth and the format of the given template. For medical records and tests task, it checks whether the generated context is diagnostically consistent with the ground truth report. If there exists any incorrectness or inconsistency in generated data, our pipeline will try to regenerate and skip to the next sample after 3 retries. Unsatisfactory generated reports are further filtered by comparing the labels of generated context and ground truth report. We use CheXpert labeler (Irvin et al., 2019), an automatic tool to extract labels of common observations from radiology reports, to extract and compare the labels of $C_i$ and $R'_i$ to ensure that no information leakage is presented in the generated context, *i.e.*, no ground truth information in generated context. Detailed prompts for quality control are shown in the Appendix.

For manual examination, we invite a group of certificated radiologists to validate the clinical correctness of the generated data, yielding a fully human-validated test set of 600 data examples, with 200 examples dedicated to each of the three sub-tasks: revision, template and medical records. The content of the other two sub-tasks, no-context and previous report, are directly used from MIMIC-CXR dataset, which does not involve any LLM-generated content, therefore no additional validation is needed. We invite five human annotators, including three junior-level radiologists with less than 5 years of medical experience, to annotate the data, and two senior-level radiologists with over 10 years of experience to review the annotations. The annotation task involved determining whether the generated data was plausible and providing a reason, a process that typically requires only entry- to mid-level experience. The medical professionals are instructed to carefully examine all information, including instructions, context, modified reports, and ground truth reports, to determine whether the entire pipeline is acceptable. Any factual errors, such as missing positive findings or hallucinated false positives, will result in rejection. However, variations in writing styles are allowed, such as treating minor conditions not mentioned as negative. Any disagreements during the annotation process were discussed to reach a consensus. The disagreement rate between annotators and reviewers regarding the correctness of the generated data was 2.7%. Annotators are also asked to rate the plausibility of each record on a scale from 1 to 10. This plausibility score is a subjective measure by medical practitioners to assess how likely the instructions or situations could occur in their daily work, ensuring that the setting aligns with real-world scenarios.

### 4.3 DATASET STATISTICS AND ANALYSIS

Using our data generation pipeline, we generate a novel dataset based on a large report generation dataset MIMIC-CXR (Johnson et al., 2019), named *MIMIC-R3G*. Since MIMIC-CXR already contains patients' previous reports, we directly use the report from dataset as ground truth and retrieved previous report as context without generation.

As noted in our quality control section, a subset of the generated dataset has been validated by certified medical professionals. The total acceptance rate is 95.5% (573 out of 600), with details illustrated in Figure 1b, including the 95% confidence intervals. The acceptance rates for the subtasks, revision, template and medical records, are 97.0%, 90.9%, and 99.5%, respectively. The overall average plausibility score for valid records is 9.58, demonstrating that the generated instructions effectively mirror daily scenarios.

Specifically, for the correction subtask, 4 errors were due to the introduction of additional modifications, and 2 errors were due to not correctly following the instructions. For the template subtask, errors were mainly due to the content being placed in the wrong template position (10 instances), such as describing a chest tube in the soft tissue section instead of the support devices section. Ad-

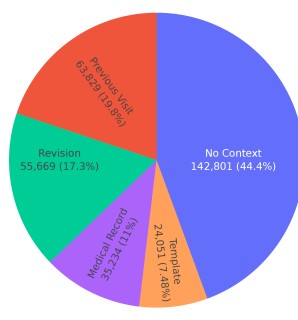
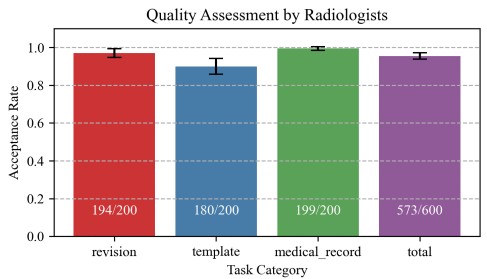

(a) Percentage of tasks        (b) Quality Control by Radiologist

Figure 1: Visualizations of *MIMIC-R3G* statistics. (a) shows the general distribution of data of different tasks. (b) shows the acceptance rate during manual quality control steps by radiologists.

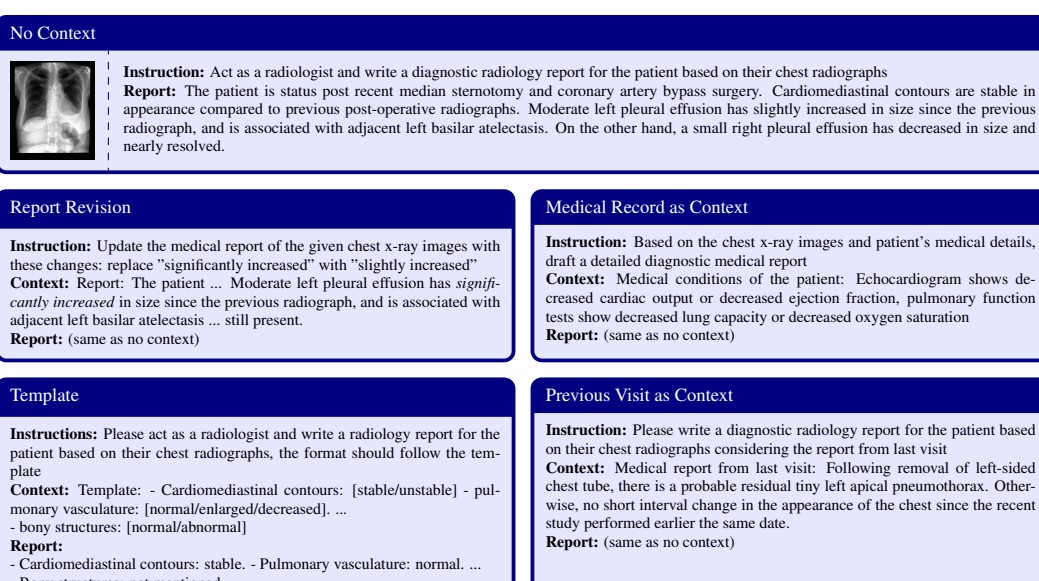

Table 1: Examples of *MIMIC-R3G* generated using the same report on different tasks

ditionally, there were 4 instances of omission and 4 of misplacement, and 2 instances of severity deviation. For the medical record subtask, 1 instance was judged unacceptable due to unclear text meaning. Considering that each record contains multiple sets of instructions and their effects, the proportion of content with errors is relatively low overall.

Among the acceptable records, those without factual errors in instruction, context, and report, 19 were marked with a plausibility score lower than 8. Of these, two were from the correction subtask, with an average score of 4.0, due to contradictions that occurred despite the text correctly following the instructions (e.g., positive cardiomegaly but normal mediastinal). 17 were from the template subtask, with an average score of 5.76, mainly due to the limited expressiveness of the templates used, making them difficult for practical clinical use, even though many of these templates were derived from RSNA templates or published studies.

In general, through careful review of the data entries by radiologists, we assessed the quality of the entire dataset generation. By retaining the qualified entries, we also delineated a higher-quality manual examined subset in the test set.

## 5 *DeMMo*: DOMAIN-ENHANCED MULTIMODAL MODEL

We propse *DeMMo*, a method tailored for instructional report generation task with context. Our objective is to train a model that given the image-text input $x = (V, I, C)$ generate output text

$y = R'$, therefore the generation process can be formalized as $p_\theta(y \mid x)$ where $\theta$ represents the model parameters to be optimized. Our model is built upon Flamingo (Alayrac et al., 2022) due to its training efficiency and good performance.

**Fusing Medical Domain Features** The general domain visual encoder of Flamingo exhibit greater diversity and generalization ability, but cannot fully capture the detailed visual feature in medical domain. Consequently, a domain specific encoder is required to capture the nuances and specific characteristics of medical images. In this paper, we employ BioViL (Boecking et al., 2022) as our medical vision encoder. As shown in Figure 2, to capitalize on the robust generalizability and expedite convergence, the original pretrained general domain visual encoder in Flamingo is still preserved in conjunction with the newly introduced medical encoder.

Specifically, given a set of images $V_i$ that contains $k$ images, the original Flamingo vision encoder outputs $n \times n$ grid features $\boldsymbol{X}_f \in \mathbb{R}^{k \times n \times n \times d_f}$, and the medical vision encoder outputs an $m \times m$ grid features $\boldsymbol{X}_m \in \mathbb{R}^{k \times m \times m \times d_m}$, where $d_f$ and $d_m$ are feature dimensions of Flamingo vision encoder and medical vision encoder, respectively. After applying a projection $\boldsymbol{W} \in \mathbb{R}^{d_m \times d_f}$ to $\boldsymbol{X}_m$ followed by flattening both grid features, we get $\boldsymbol{X}_f \in \mathbb{R}^{kn^2 \times d_f}$ and $\boldsymbol{X}_m \in \mathbb{R}^{km^2 \times d_f}$. We adopt the idea of LLaMA-Adapter (Zhang et al., 2023) to insert a learnable adaption prompt $\boldsymbol{P}_l \in \mathbb{R}^{m^2 \times d_f}$ into the perceiver resampler independently for each layer $l$. Each flattened feature from medical vision encoder is then added element-wise to $\boldsymbol{P}_l$ to form the medical visual feature prepared for attention. Similar to vanilla Flamingo, a predefined number of latent queries are cross-attended to the concatenation of queries and visual features. Formally, denote $t$ as the number of latent queries. At layer $l$, $\boldsymbol{Q}_l \in \mathbb{R}^{t \times d_f}$ is the latent queries, and $\boldsymbol{V}_l = \boldsymbol{K}_l \in \mathbb{R}^{(km^2 + kn^2 + t) \times d_f}$ is the concatenation of medical visual features, original visual features from Flamingo vision encoder, and the latent queries. Then, the similarity scores are computed as

$$S_l = \left( \boldsymbol{Q}_l \boldsymbol{W}_l^Q \right) \left( \boldsymbol{K}_l \boldsymbol{W}_l^K \right)^\top / \sqrt{d_h} \in \mathbb{R}^{t \times (km^2 + kn^2 + t)} \tag{1}$$

where $\boldsymbol{W}_l^Q, \boldsymbol{W}_l^K \in \mathbb{R}^{d_f \times d_h}$ are query and key projections respectively at layer $l$, and $d_h$ represents the hidden feature dimension.

After obtaining the similarity scores, to ensure that no instability will be introduced when initializing the model with medical feature introduced, we follow (Zhang et al., 2023) to apply softmax independent on two splits of the similarity score matrix, one on the scores corresponding to the Flamingo visual features and latent queries, and the other one on the scores corresponding to the newly introduced medical visual features. Specifically, $S_l$ could be separated into:

$$S_l = \left[ S_l^m; S_l^f; S_l^q \right] \tag{2}$$

where $S_l^m \in \mathbb{R}^{t \times km^2}, S_l^f \in \mathbb{R}^{t \times kn^2}, S_l^q \in \mathbb{R}^{t \times t}$ represent similarity scores of the queries with respect to medical features, Flamingo vision encoder features, and the latent queries, respectively. We then apply a $\tanh$ gate controlled by a zero-initialized trainable parameter $g_l$. The resulting attention score at layer $l$ is:

$$\text{Attn}_l = \left[ \tanh(g_l) \cdot \text{Softmax}(S_l^m); \text{Softmax}\left( \left[ S_l^f; S_l^q \right] \right) \right]$$

In this way, when the model is initialized, medical visual features will have zero effect, and the forward process is equivalent to the forward process of a pretrained vanilla Flamingo. As the training advances, the gate parameter $g_l$ will be updated to gradually introduce the influence from medical visual features.

**Pathological Guidance** We further introduce the detailed implementation of philological guidance. Specifically, given a chest medical image and a pathology phrase, BioViL (Boecking et al., 2022) is able to output a heatmap on the image associated with the phrase. In our training phase, we apply the CheXpert labels extracted from the ground truth report to find maxima on the heatmap and crop a zoomed in region of interest for each image. We proceed by concatenating the zoomed in regions of interest with the original images as the input fed into the perceiver resampler. Additionally, to enable this guidance during inference when ground truth labels are not available, we augment the perceiver resampler with binary classifiers for each pathology category, which imposes additional constraints to ensure that the latent query output of the perceiver contains pathology categorization

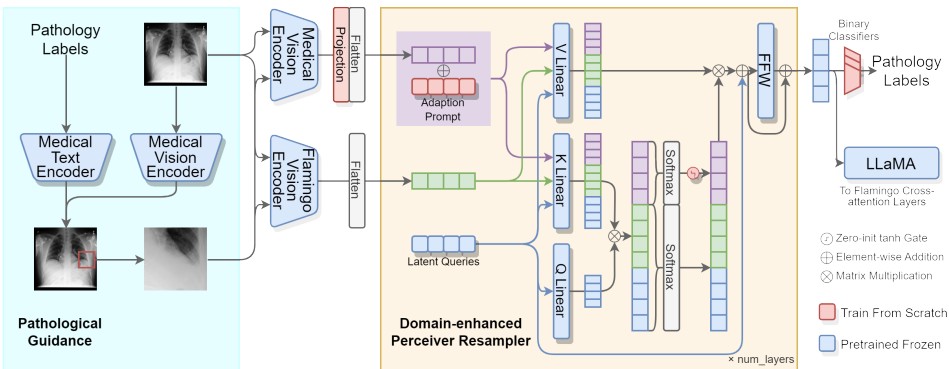

Figure 2: Architecture of *DeMMo*

information. During inference, the medical image is initially passed through the perceiver resampler only to obtain the corresponding pathology label. Subsequently, this label is utilized to extract zoomed-in region of interest from the original image with BioViL (Boecking et al., 2022) . Finally, the extracted region along with the full medical images undergoes a full forward pass through the entire pipeline.

The rest are same as vanilla Flamingo model, where the attended queries pass through another feed-forward network before next layer, and the last perceiver layer output is inserted into Flamingo Cross-attention layers. We only tune the medical vision encoder projection, adaption prompts, zero-initialized gates, and the binary disease classifiers, along with the Flamingo cross-attention layer in LLaMA. Our fine-tuning approach ensures a seamless integration of domain-specific knowledge without introducing instability in the model initialization thus compromising the generalizability of the model.

## 6 EXPERIMENTS

### 6.1 EXPERIMENTAL SETTINGS

Following (Chen et al., 2020; 2021; Wang et al., 2022a; Nicolson et al., 2023), we use the samples with findings section and at least one frontal view images in MIMIC-CXR to generate our dataset and conduct experiment. This results in the dataset statistics shown in Figure 1a. *MIMIC-R3G* contains two test datasets. *MIMIC-R3G*-test-A is the validated dataset containing 3,846 samples, where 3,246 samples are from the no context generation task and previous visit as context generation task, which are directly referenced from the original MIMIC-CXR dataset, and the remaining 600 samples are generated and human-validated across three tasks: template, revision, and medical record as context. This test set is intended to include all data samples that we are certain are correct. *MIMIC-R3G*-test-B is the full test set containing all 8,965 generated test samples, where part of the samples in revision, template and medical record sub-tasks have not been manually validated. Due to page limit, in this section we only report the benchmark results on *MIMIC-R3G*-test-A, and the results on *MIMIC-R3G*-test-B is presented in Section D of the Appendix. We adopt natural language generation (NLG) metrics that measures text similarity between generated and ground truth report, including BLEU (B@n) (Papineni et al., 2002), METEOR (M) (Banerjee & Lavie, 2005), and ROUGE-L (R-L) (Lin, 2004). Following previous works, we also utilize CheXpert, an automatic labeling pipeline to extract observation labels from chest X-ray reports, to evaluate clinical efficacy (CE) in terms of micro-averaged label precision (P), recall (R), and F1-score (F1). More detailed experimental settings on model implementations and hyper-parameters are introduced in the Appendix.

### 6.2 PERFORMANCE BENCHMARK ON *MIMIC-R3G*

**Compared Baselines.** We experiment multiple open-sourced general and medical domain text-image models that may be suitable for report generation tasks on the fully-validated test set of *MIMIC-R3G*. We include results for ChatCAD+ Zhao et al. (2023), GPT-4V OpenAI (2023), Med-Flamingo Moor et al. (2023), LLaVa-Med Li et al. (2023b), RadFM Wu et al. (2023), LLM-CXR Lee et al. (2023), CvT2DistilGPT2 Nicolson et al. (2023), and Flamingo Alayrac et al. (2022); Awadalla et al. (2023). The Flamingo model is fine-tuned on our training dataset.

| Task | Method | B@1 | B@2 | B@3 | B@4 | M | R-L | P | R | F1 |
|---|---|---|---|---|---|---|---|---|---|---|
| No Context | Cvt2DistilGPT2 | 0.299 | 0.188 | 0.127 | 0.091 | 0.260 | **0.249** | **0.538** | 0.421 | 0.472 |
| | ChatCAD+ | 0.307 | 0.160 | 0.088 | 0.052 | 0.266 | 0.189 | 0.335 | **0.613** | 0.433 |
| | GPT-4V | 0.126 | 0.063 | 0.030 | 0.015 | 0.240 | 0.121 | 0.368 | 0.405 | 0.385 |
| | Med-Flamingo | 0.092 | 0.026 | 0.009 | 0.004 | 0.071 | 0.054 | 0.159 | 0.084 | 0.110 |
| | LLaVa-Med | 0.076 | 0.025 | 0.008 | 0.002 | 0.082 | 0.114 | 0.220 | 0.096 | 0.134 |
| | RadFM | 0.111 | 0.061 | 0.037 | 0.024 | 0.126 | 0.135 | 0.332 | 0.224 | 0.268 |
| | LLM-CXR | 0.071 | 0.032 | 0.017 | 0.009 | 0.093 | 0.097 | 0.377 | 0.270 | 0.310 |
| | Flamingo* | 0.365 | 0.219 | 0.139 | 0.097 | 0.285 | 0.231 | 0.438 | 0.411 | 0.424 |
| | Ours | **0.375** | **0.227** | **0.146** | **0.103** | **0.296** | 0.242 | 0.500 | 0.461 | **0.480** |
| Revision | Cvt2DistilGPT2 | 0.292 | 0.177 | 0.115 | 0.080 | 0.248 | 0.234 | 0.520 | 0.402 | 0.453 |
| | ChatCAD+ | 0.636 | 0.570 | 0.521 | 0.480 | 0.710 | 0.647 | 0.868 | 0.846 | 0.857 |
| | GPT-4V | 0.518 | 0.441 | 0.382 | 0.335 | 0.710 | 0.620 | 0.853 | 0.863 | 0.858 |
| | Med-Flamingo | 0.303 | 0.228 | 0.183 | 0.150 | 0.408 | 0.304 | 0.560 | 0.596 | 0.577 |
| | LLaVa-Med | 0.385 | 0.276 | 0.214 | 0.172 | 0.405 | 0.316 | 0.569 | 0.538 | 0.553 |
| | RadFM | 0.049 | 0.030 | 0.021 | 0.016 | 0.077 | 0.074 | 0.350 | 0.122 | 0.164 |
| | LLM-CXR | 0.183 | 0.118 | 0.085 | 0.064 | 0.189 | 0.201 | 0.488 | 0.356 | 0.412 |
| | Flamingo* | 0.737 | 0.687 | 0.648 | 0.615 | 0.765 | 0.759 | 0.884 | 0.811 | 0.847 |
| | Ours | **0.837** | **0.790** | **0.752** | **0.719** | **0.832** | **0.826** | **0.934** | **0.879** | **0.898** |
| Template | Cvt2DistilGPT2 | 0.111 | 0.061 | 0.036 | 0.024 | 0.139 | 0.159 | 0.591 | 0.327 | 0.421 |
| | ChatCAD+ | 0.515 | 0.445 | 0.397 | 0.358 | 0.454 | 0.416 | 0.507 | 0.521 | 0.514 |
| | GPT-4V | 0.308 | 0.244 | 0.202 | 0.171 | 0.406 | 0.330 | 0.583 | 0.509 | 0.543 |
| | Med-Flamingo | 0.153 | 0.076 | 0.046 | 0.036 | 0.093 | 0.108 | 0.263 | 0.129 | 0.173 |
| | LLaVa-Med | 0.158 | 0.090 | 0.063 | 0.049 | 0.121 | 0.121 | 0.449 | 0.204 | 0.280 |
| | RadFM | 0.080 | 0.039 | 0.021 | 0.012 | 0.079 | 0.063 | 0.280 | 0.118 | 0.166 |
| | LLM-CXR | 0.028 | 0.011 | 0.005 | 0.002 | 0.064 | 0.082 | 0.414 | 0.204 | 0.273 |
| | Flamingo* | 0.469 | 0.402 | 0.348 | 0.350 | 0.443 | 0.447 | 0.577 | 0.449 | 0.505 |
| | Ours | **0.534** | **0.461** | **0.409** | **0.367** | **0.533** | **0.483** | **0.684** | **0.564** | **0.618** |
| Previous Report | Cvt2DistilGPT2 | 0.306 | 0.192 | 0.129 | 0.092 | 0.262 | **0.250** | **0.526** | 0.412 | 0.462 |
| | ChatCAD+ | 0.310 | 0.168 | 0.100 | 0.063 | **0.290** | 0.199 | 0.511 | 0.523 | **0.516** |
| | GPT-4V | 0.166 | 0.088 | 0.046 | 0.026 | 0.281 | 0.159 | 0.435 | **0.589** | 0.500 |
| | Med-Flamingo | 0.164 | 0.077 | 0.042 | 0.026 | 0.177 | 0.133 | 0.447 | 0.333 | 0.382 |
| | LLaVa-Med | 0.271 | 0.131 | 0.072 | 0.044 | 0.215 | 0.159 | 0.433 | 0.304 | 0.357 |
| | RadFM | 0.167 | 0.087 | 0.050 | 0.031 | 0.144 | 0.131 | 0.463 | 0.328 | 0.384 |
| | LLM-CXR | 0.075 | 0.036 | 0.020 | 0.012 | 0.103 | 0.113 | 0.431 | 0.295 | 0.369 |
| | Flamingo* | 0.356 | 0.214 | 0.135 | 0.094 | 0.281 | 0.229 | 0.438 | 0.366 | 0.399 |
| | Ours | **0.383** | **0.231** | **0.147** | **0.098** | 0.287 | 0.242 | 0.511 | 0.493 | 0.502 |
| Medical Record | Cvt2DistilGPT2 | 0.306 | 0.191 | 0.126 | 0.089 | 0.267 | 0.257 | 0.566 | 0.418 | 0.481 |
| | ChatCAD+ | 0.177 | 0.089 | 0.051 | 0.032 | 0.240 | 0.128 | 0.447 | 0.598 | 0.512 |
| | GPT-4V | 0.095 | 0.051 | 0.028 | 0.017 | 0.235 | 0.103 | 0.423 | **0.656** | 0.514 |
| | Med-Flamingo | 0.168 | 0.080 | 0.047 | 0.030 | 0.179 | 0.132 | 0.518 | 0.494 | 0.506 |
| | LLaVa-Med | 0.238 | 0.114 | 0.064 | 0.040 | 0.213 | 0.147 | 0.517 | 0.484 | 0.499 |
| | RadFM | 0.133 | 0.054 | 0.028 | 0.016 | 0.086 | 0.084 | 0.422 | 0.285 | 0.340 |
| | LLM-CXR | 0.114 | 0.056 | 0.030 | 0.017 | 0.120 | 0.120 | 0.551 | 0.389 | 0.456 |
| | Flamingo* | 0.374 | 0.245 | 0.171 | 0.128 | 0.321 | 0.271 | 0.560 | 0.464 | 0.508 |
| | Ours | **0.394** | **0.269** | **0.195** | **0.150** | **0.345** | **0.302** | **0.574** | 0.518 | **0.544** |
| Average | Cvt2DistilGPT2 | 0.263 | 0.162 | 0.107 | 0.075 | 0.235 | 0.230 | 0.548 | 0.396 | 0.458 |
| | ChatCAD+ | 0.389 | 0.286 | 0.231 | 0.197 | 0.392 | 0.316 | 0.533 | **0.620** | 0.566 |
| | GPT-4V | 0.243 | 0.177 | 0.138 | 0.113 | 0.374 | 0.267 | 0.532 | 0.604 | 0.560 |
| | Med-Flamingo | 0.176 | 0.097 | 0.065 | 0.049 | 0.186 | 0.147 | 0.389 | 0.327 | 0.350 |
| | LLaVa-Med | 0.226 | 0.127 | 0.084 | 0.061 | 0.027 | 0.171 | 0.438 | 0.325 | 0.365 |
| | RadFM | 0.108 | 0.054 | 0.031 | 0.020 | 0.102 | 0.097 | 0.369 | 0.215 | 0.264 |
| | LLM-CXR | 0.094 | 0.051 | 0.031 | 0.021 | 0.114 | 0.123 | 0.452 | 0.303 | 0.364 |
| | Flamingo* | 0.460 | 0.353 | 0.288 | 0.257 | 0.419 | 0.387 | 0.579 | 0.500 | 0.537 |
| | Ours | **0.505** | **0.396** | **0.330** | **0.287** | **0.459** | **0.419** | **0.641** | 0.583 | **0.608** |

Table 2: Comparison of our model with other baselines on *MIMIC-R3G*-test-A. B@n, M, R-L represent the NLG metrics BLEU, METEOR, and ROUGE-L respectively. P, R, F1 represent the CE metrics CheXpert precision, recall, and F1-score respectively. Flamingo* represents Flamingo model finetuned on training set of *MIMIC-R3G*.

Table 2 presents the performance of *DeMMo* and aforementioned methods on each of the *MIMIC-R3G* sub-tasks, respectively. Note that the test splits of each sub-task are not identical and hence the performance is not comparable across different tasks. We observe that conventional encode-decoder architecture report generation model like CvT2DistilGPT2 achieves decent performance on CE and NLG metrics, but falls short in revision and template tasks. This limitation can be attributed

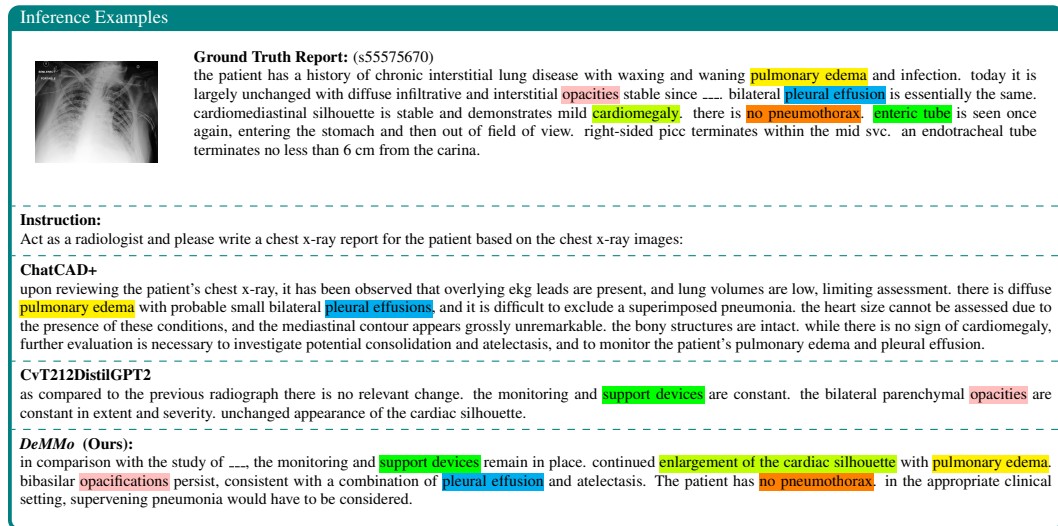

Table 3: A inference example by *DeMMo* and the other compared methods.

to the model's original training data, which is solely MIMIC-CXR with no instructional contexts, leading it to ignore any textual inputs during inferences. Training-free generation pipelines utilizing general-domain LLM such as ChatCAD+ and GPT-4V exhibit strong performance across various tasks in terms of CE metrics, demonstrating their adeptness at contextual understanding. Chat-CAD+, enhanced by a pretrained disease classifier, performs even better in generating precise diagnosis. However, these models often generate verbose outputs and are susceptible to hallucination, which adversely affects their NLG scores. Additionally, their tendency to enumerate all conceivable diseases leads to exceptionally high recall, at the expense of precision. Multimodal LLMs that have been fine-tuned on medical domain data, such as Med-Flamingo, LLaVa-Med, and RadFM, tend to show low performance on many tasks as well. This is predominantly because their training datasets are composed mostly of medical visual question answering data, which skews towards brief and succinct responses. Consequently, these models struggle to adhere to instructions that require the generation of detailed and comprehensive reports. Multimodal LLM fine-tuned on our *MIMIC-R3G* training set (Flamingo*) achieves promising results on both NLG and CE metrics on all tasks, underscores the efficacy of our generated context data in enhancing these instructional report generation tasks. Moreover, our newly proposed model, *DeMMo*, exhibits further enhancements, achieving highest scores in NLG and CE metrics on most tasks, which highlights the effectiveness of our novel design in adapting general-domain multimodal LLM for report generation tasks that involve instructional contexts. Table 3 shows an example output by *DeMMo* and other comparison methods.

## 7 CONCLUSIONS

In this paper, we propose a highly interactive real-world radiology report generation problem setting (R3G). R3G requires models to be highly interactive, to follow instructions and consider various context information. A new benchmark dataset for the real-world report generation is built with a unified data generation pipeline. A novel Domain-enhanced Multi-Modal (*DeMMo*) model is proposed to enhance the medical domain specific ability of conventional LLM. Experiments demonstrate that *DeMMo* attains competitive performance across all real-world tasks.

## 8 DATA AVAILABILITY

This dataset is derived from MIMIC-CXR, so users are required to sign the MIMIC-CXR Data Use Agreement (DUA) and download MIMIC-CXR through PhysioNet to use it with this dataset. MIMIC-R3G source data will be released on PhysioNet, along with the official dataset documentation and annotation requirements. Each data sample includes the instruction and context, the ground truth report, and the image IDs of the corresponding X-ray images. Source code for generating context along with all prompts used, and source code for compiling the generated text into JSON format dataset, will be made available through GitHub.

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

## A  MORE RELATED WORKS

**Multimodal LLMs** With the remarkable success of LLMs, researchers started to explore the possibility to integrate visual modality into LLMs for various visual-language tasks. Early works such as BLIP-2 (Li et al., 2023c) leveraged a query Transformer to connect visual features to LLM. Flamingo (Alayrac et al., 2022) introduced extra trainable layers within LLM in addition to a Transformer to bridge visual and language modalities. LLaVa (Liu et al., 2023) and MIMIC-IT (Li et al., 2023a) leveraged GPT/ChatGPT to build visual instruction tuning datasets and developed multimodal LLMs as general instruction-follow visual agents. Following their ideas, we construct a real-world report generation dataset by building a unified data generation pipeline leveraging ChatGPT.

**Medical LLMs** Numerous works have applied LLM within the medical domain through fintuning a general domain LLM. Med-PaLM (Singhal et al., 2022) and Med-PaLM 2 (Singhal et al., 2023) are medical domain-specific language models developed through instruction fine-tuning based on general domain LLMs. Med-PaLM M (Tu et al., 2023) further fine-tunes PaLM-E to the medical domain using multimodal medical data for medical vision-language tasks. LLaVa-Med (Li et al., 2023b) and Med-Flamingo (Moor et al., 2023) similarly fine-tune their general domain base models using domain-specific data to enhance medical question-answering and conversational capabilities. Different from fine-tuning LLMs for medical domain, ChatCAD (Wang et al., 2023) and ChatCAD+ (Zhao et al., 2023) interact with users by connecting medical domain models with ChatGPT via language prompts. We observe that this framework is capable of doing all of our proposed real-world report generation tasks without training using extra task-specific data.

## B  *MIMIC-R3G* GENERATION AND VALIDATION EXAMPLES

---

**Example of Generating and Validating Report Revision Data**

**System Message:** You are a radiologist and will intentionally make some diagnostic errors in a chest X-ray radiology report. You will be given a correct findings section of the chest X-ray report, and you will be asked to rewrite the report with a few, around 2-4, diagnostic errors. Provide the rewritten, incorrect report only.

**User (GT report):** PA and lateral views of the chest provided. The lungs are adequately aerated. There is a focal consolidation at the left lung base adjacent to the lateral hemidiaphragm. There is mild vascular engorgement. There is bilateral apical pleural thickening. The cardio mediastinal silhouette is remarkable for aortic arch calcifications. The heart is top normal in size.

**Response:** PA and lateral views of the chest provided. The lungs are adequately aerated. There is no focal consolidation. There is mild vascular engorgement. There is no bilateral apical pleural thickening. The cardio mediastinal silhouette is remarkable for aortic arch calcifications. The heart is mildly enlarged in size.

- - - - - - - - - - - - - - - - - - - - - - - - - - - - - - - - - - - - -

**System Message:** You are a radiologist who helps check the correctness of chest x-ray radiology report. Given a correct report and an incorrect report, provide 2 to 3 short and concise pointers on how to fix the incorrect report into the correct report. Do not be verbose or verbatim. List the instructions or suggestions only. The doctor has access to the radiograph, therefore your instructions could be vague or only state what the problems are without actual instruction to fix them.

**User:** Incorrect report: ⟨previous response⟩ Correct report: ⟨GT report⟩

**Response:** 1. Add focal consolidation. 2. The patient has bilateral apical pleural thickening.

- - - - - - - - - - - - - - - - - - - - - - - - - - - - - - - - - - - - -

**System Message:** You are a radiologist who helps check the correctness of radiology reports. Given a pair of correct and incorrect reports and the instructions to revise the incorrect report, print "yes" if the instruction can be used to modify the incorrect report into correct report, otherwise print "no". Specifically pay attention to the revision direction so that the instructions do not modify correct report into incorrect ones.

**User:** Incorrect report: ⟨text in first response⟩ Correct report: ⟨GT report⟩ Instructions: ⟨text in second response⟩

**Response:** yes

- - - - - - - - - - - - - - - - - - - - - - - - - - - - - - - - - - - - -

$I_i = \langle$ Text in second response$\rangle$
$C_i = \langle$ Text in first response$\rangle$
$R'_i = \langle$ GT report$\rangle$

---

**Example of Generating and Validating Template Data**

**System Message:** Template:
REPORT: [Imaging Protocol]
COMPARISON: [None/Compare with former image]
FINDINGS:
        Lines/tubes:
        Lungs:
        Pleura:
        Heart and mediastinum:
        Bones:
        Support Devices:
You are an assistant who helps format radiology reports using structured templates. You will be given a free-text radiology report, and you need to rewrite the report using this template by fill in the blanks. Leave any section blank if the report doesn't mention related information.

**User (GT Report):** CHEST (PORTABLE AP) INDICATION: ___ year old woman with pleurx for malignant effusion // ? appropriate position, response. please perform by 7 AM. FINDINGS: As compared to the previous image, no relevant change is seen. The extent of the left pleural effusion and the position of the

---

left pleural drain are constant. Constant appearance of the mild opacity at the right lung bases. No new parenchymal changes. Unchanged size of the cardiac silhouette.
**Response:**
REPORT: CHEST (PORTABLE AP)
COMPARISON: Compare with former image
FINDINGS:
    Lines/tubes:
    Lungs: Constant appearance of the mild opacity at the right lung bases. No new parenchymal changes.
    Pleura: The extent of the left pleural effusion and the position of the left pleural drain are constant.
    Heart and mediastinum: Unchanged size of the cardiac silhouette.
    Bones:
    Support Devices:

---

**System Message:** You are a radiologist's assistant who helps check the consistency between a free-text report and a templated report. Output "yes" if the diagnosis of the templated report matches the diagnosis in the free-text report on all pathologies, otherwise output "no" if there are any errors or inconsistencies.
**User:** Free-text Report: ⟨GT report⟩ Templated Report:⟨text in first response⟩
**Response:** yes

---

**System Message:** You are a radiologist's assistant who helps check the consistency between a template and a formatted report. Output "yes" if the report follows the format of the given template, otherwise output "no" if the report does not follow the structure of the template.
**User:** Template: ⟨template⟩ Templated Report:⟨text in first response⟩
**Response:** yes

---

$I_i = \langle$Manually designed instructions$\rangle$
$C_i = \langle$Template$\rangle$
$R'_i = \langle$Text in first response$\rangle$

---

### Example of Generating and Validating Medical Record Data

**System Message:** You are a radiology assistant who help extract information from a chest x-ray radiology report. Output the corresponding section related to indication or medical history, if there are no such sections, output "none".
**User (GT Report):** FINAL REPORT INDICATION: ___-year-old female with fall, weakness. Please assess for pneumonia. TECHNIQUE: AP frontal and lateral chest radiographs were obtained. COMPARISON: None available. FINDINGS: Lung volumes are low. No focal consolidation, pleural effusion or pneumothorax is seen. There is no pulmonary edema. The cardiac silhouette is accentuated by low lung volumes and AP projection. IMPRESSION: Low lung volumes.
**Response:** ___-year-old female with fall, weakness. Please assess for pneumonia.

---

**System Message:** You are a clinical assistant that helps radiologists retrieve information of patient's other medical records and examination results. You will be given a chest x-ray diagnosis report, and you need to infer the patient's possible medical conditions or history based on the indication and/or impression. Only list 2 to 4 of the possible abnormal medical history or conditions that are related to the indication, findings or impression of the report. Be short and concise and only output the list.
**User:** ⟨GT report⟩
**Response:**
1. Chronic obstructive pulmonary disease (COPD)
2. Asthma
3. Muscular dystrophy
4. Myasthenia gravis

---

**System Message:** You are a clinical assistant that helps radiologists retrieve information of patient's other medical records and examination results. You will be given a chest x-ray diagnosis report, and you need to infer possible medical lab tests that the patient should undergo. Only list 2 to 4 of the possible medical tests that are related to the indication, findings, or impression of the report. Be short and concise and only output the list.
**User:** ⟨GT report⟩
**Response:**
1. Pulmonary Function Test
2. Complete Blood Count (CBC)
3. Arterial Blood Gas Analysis

---

**System Message:** You are a clinical assistant that helps radiologists retrieve information of patient's other medical records and examination results. You will be given a chest x-ray diagnosis report with names of some medical tests, and you need to infer the reasonable test results and/or values that the patient should have based on the report indication, findings, and impression. Be concise and definitive and do not use uncertain terms. Only output the list without any other statement.
**User:** Report: ⟨GT report⟩ Tests: ⟨text in previous response⟩
**Response:**
1. Pulmonary Function Test: Reduced total lung capacity, reduced vital capacity, reduced forced vital capacity.
2. Complete Blood Count (CBC): Normal range.
3. Arterial Blood Gas Analysis: Normal pH, normal PaCO2, normal PaO2, normal HCO3.

---

**System Message:** You are a clinical assistant that helps radiologists check information consistency of patient's report and medical record. You will be given a chest x-ray diagnosis report, along with some patient's medical history, condition, and lab test results. You need to check whether the medical history/condition and lab test results are diagnostically consistent with the radiology report with no factual errors. If they are not consistent or have any factual errors, output "no". If the report and other information are correct and consistent, output "yes".
**User:** Indication: ⟨text in first response⟩ Medical history: ⟨text in second response⟩ Medical test: ⟨text in forth response⟩ Report: ⟨GT report⟩
**Response:** yes

---

$I_i = \langle$Manually designed instructions$\rangle$
$C_i = \langle$Text in first, second, and forth response$\rangle$
$R'_i = \langle$GT report$\rangle$

## C    MORE ON EXPERIMENTS

### C.1    DATASETS AND IMPLEMENTATION DETAILS

The proposed generated benchmark datasets are built upon the ground-truth report in MIMIC-CXR, which is the largest widely used report generation dataset. It consists of chest X-ray radiographs and

| Method | B@1 | B@2 | B@3 | B@4 | M | P | R | F1 |
|---|---|---|---|---|---|---|---|---|
| R2Gen (Chen et al., 2020) | 0.353 | 0.218 | 0.145 | 0.103 | 0.142 | 0.333 | 0.273 | 0.276 |
| CMN (Chen et al., 2021) | 0.353 | 0.218 | 0.148 | 0.106 | 0.142 | 0.334 | 0.275 | 0.278 |
| XPRONET (Wang et al., 2022a) | 0.344 | 0.215 | 0.146 | 0.105 | 0.138 | - | - | - |
| CvT2DistilGPT2 (Nicolson et al., 2023) | **0.393** | **0.248** | **0.171** | **0.127** | 0.155 | 0.367 | 0.418 | 0.390 |
| *DeMMo* (Ours) | 0.375 | 0.227 | 0.146 | 0.103 | **0.296** | **0.500** | **0.461** | **0.480** |

Table 4: Comparison of *DeMMo* with conventional report generation methods. The highest and the second highest performance are highlighted in bold and underline respectively.

| Task | Metrics | w/o Medical Encoder | w/o General Encoder | w/o pathological guidance | *DeMMo* |
|---|---|---|---|---|---|
| No Context | BLEU@1 | 0.365 | **0.376** | 0.373 | 0.375 |
| | Precision | 0.438 | 0.487 | 0.491 | **0.500** |
| | Recall | 0.411 | 0.453 | 0.451 | **0.461** |
| | F1 Score | 0.424 | 0.469 | 0.470 | **0.480** |
| Revision | BLEU@1 | 0.737 | 0.747 | 0.777 | **0.837** |
| | Precision | 0.884 | 0.894 | 0.845 | **0.934** |
| | Recall | 0.811 | 0.818 | 0.817 | **0.879** |
| | F1 Score | 0.847 | 0.854 | 0.831 | **0.898** |
| Template | BLEU@1 | 0.469 | 0.429 | 0.529 | **0.534** |
| | Precision | 0.577 | 0.659 | 0.683 | **0.684** |
| | Recall | 0.449 | 0.489 | 0.530 | **0.564** |
| | F1 Score | 0.505 | 0.561 | 0.597 | **0.618** |
| Previous Report | BLEU@1 | 0.356 | 0.357 | 0.370 | **0.383** |
| | Precision | 0.438 | 0.500 | 0.503 | **0.511** |
| | Recall | 0.366 | 0.421 | 0.436 | **0.493** |
| | F1 Score | 0.399 | 0.457 | 0.467 | **0.502** |
| Medical Record | BLEU@1 | 0.374 | 0.382 | 0.381 | **0.394** |
| | Precision | 0.560 | 0.573 | 0.580 | **0.574** |
| | Recall | 0.464 | 0.437 | 0.446 | **0.518** |
| | F1 score | 0.508 | 0.496 | 0.504 | **0.544** |

Table 5: Ablation studies on the performance comparison of different components in *DeMMo*, including medical encoder, general Flamingo encoder, and pathological guidance.

reports of 227,835 studies from 64,588 patients, with a total of 227,835 reports and 377,110 x-ray images. The official training and test splits of MIMIC-CXR includes 386,960 images and 222,758 reports in training set and 5159 images and 3269 reports in test set.

We adopt OpenFlamingo (Awadalla et al., 2023), which is an opensource implementation of the Flamingo architecture. We use BioViL (Boecking et al., 2022) as our medical vision encoder. The BioViL medical vision encoder outputs a $15 \times 15$ grid of features with feature dimension 2048, which is then flattened and projected into 225 1024-dimensional vectors, which is same as the feature dimension of original CLIP ViT-L/14 encoder in Flamingo. The length of adaption prompt in perceiver sampler is same as the number of visual features from medical vision encoder output, which is 225. We maintain other model design parameters, *e.g.*, hidden dimension and number of attention heads, consistent with the OpenFlamingo implementation. For each data sample, we randomly sample two frontal view chest x-ray images associated with the study, or add a dummy zero-valued image if there is only one available frontal view image. We train our model and vanilla Flamingo on *MIMIC-R3G* data for 10 epochs with 2 batch size in all experiments. We use an ADAMW optimizer with $\beta_1 = 0.9, \beta_2 = 0.999$ and weight decay of 0.01 and set the learning rate 1e-4 with a 1000-step warm-up and a cosine decay schedule. Beam search with beam size of 3 is used for report generation. We train the model on 1 80G A100 GPU.

## C.2 Performance Comparison on Conventional Report Generation

To show the efficacy of our model architecture design, we also evaluated the performance of *DeMMo* on conventional report generation task without generated context. Specifically, we train *DeMMo* using the original MIMIC-CXR dataset to compare with other conventional report generation models under the same setting. For a fair comparison, only generation methods that do not use extra medical dataset, knowledge graphs, or disease label or image classifier are compared. The

performance of the comparison methods are directly cited from papers. As shown in Table 4, our methods significantly outperform existing conventional report generation methods in terms of CE metrics and a comparable performance in terms of NLG metrics.

## C.3 PERFORMANCE ON *MIMIC-R3G*-TEST-B

We test various methods on the generated, not fully validated test set. Results are shown in Table 6

| Task | Method | B@1 | B@2 | B@3 | B@4 | M | R-L | P | R | F1 |
|---|---|---|---|---|---|---|---|---|---|---|
| No Context | Cvt2DistilGPT2 | 0.299 | 0.188 | 0.127 | 0.091 | 0.260 | **0.249** | **0.538** | 0.421 | 0.472 |
| | ChatCAD+ | 0.307 | 0.160 | 0.088 | 0.052 | 0.266 | 0.189 | 0.335 | **0.613** | 0.433 |
| | GPT-4V | 0.126 | 0.063 | 0.030 | 0.015 | 0.240 | 0.121 | 0.368 | 0.405 | 0.385 |
| | Med-Flamingo | 0.092 | 0.026 | 0.009 | 0.004 | 0.071 | 0.054 | 0.159 | 0.084 | 0.110 |
| | LLaVa-Med | 0.076 | 0.025 | 0.008 | 0.002 | 0.082 | 0.114 | 0.220 | 0.096 | 0.134 |
| | RadFM | 0.111 | 0.061 | 0.037 | 0.024 | 0.126 | 0.135 | 0.332 | 0.224 | 0.268 |
| | Flamingo* | 0.365 | 0.219 | 0.139 | 0.097 | 0.285 | 0.231 | 0.438 | 0.411 | 0.424 |
| | Ours | **0.375** | **0.227** | **0.146** | **0.103** | **0.296** | 0.242 | 0.500 | 0.461 | **0.480** |
| Revision | Cvt2DistilGPT2 | 0.302 | 0.188 | 0.126 | 0.090 | 0.260 | 0.247 | 0.536 | 0.428 | 0.476 |
| | ChatCAD+ | 0.639 | 0.571 | 0.521 | 0.479 | 0.719 | 0.655 | 0.860 | 0.866 | 0.863 |
| | GPT-4V | 0.514 | 0.435 | 0.376 | 0.330 | 0.707 | 0.617 | 0.821 | **0.907** | 0.862 |
| | Med-Flamingo | 0.294 | 0.221 | 0.177 | 0.145 | 0.414 | 0.307 | 0.580 | 0.626 | 0.601 |
| | LLaVa-Med | 0.379 | 0.270 | 0.208 | 0.167 | 0.411 | 0.318 | 0.572 | 0.562 | 0.567 |
| | RadFM | 0.048 | 0.030 | 0.021 | 0.016 | 0.074 | 0.067 | 0.228 | 0.112 | 0.150 |
| | Flamingo* | 0.774 | **0.643** | **0.618** | 0.596 | **0.770** | **0.762** | 0.848 | 0.815 | 0.831 |
| | Ours | **0.784** | 0.686 | 0.641 | **0.630** | 0.740 | 0.726 | **0.894** | 0.837 | **0.865** |
| Template | Cvt2DistilGPT2 | 0.116 | 0.063 | 0.038 | 0.025 | 0.140 | 0.155 | 0.574 | 0.316 | 0.407 |
| | ChatCAD+ | 0.506 | 0.433 | 0.381 | 0.340 | 0.443 | 0.409 | 0.553 | **0.572** | 0.562 |
| | GPT-4V | 0.326 | 0.255 | 0.210 | 0.177 | 0.410 | 0.331 | 0.599 | 0.485 | 0.536 |
| | Med-Flamingo | 0.155 | 0.081 | 0.053 | 0.039 | 0.092 | 0.104 | 0.296 | 0.128 | 0.179 |
| | LLaVa-Med | 0.168 | 0.094 | 0.064 | 0.048 | 0.126 | 0.124 | 0.457 | 0.218 | 0.295 |
| | RadFM | 0.084 | 0.041 | 0.023 | 0.014 | 0.079 | 0.063 | 0.279 | 0.113 | 0.161 |
| | Flamingo* | 0.470 | 0.407 | 0.362 | 0.326 | 0.447 | 0.413 | 0.572 | 0.440 | 0.497 |
| | Ours | **0.530** | **0.449** | **0.409** | **0.366** | **0.535** | **0.480** | **0.660** | 0.561 | **0.583** |
| Previous Report | Cvt2DistilGPT2 | 0.306 | 0.192 | 0.129 | 0.092 | 0.262 | **0.250** | 0.526 | 0.412 | 0.462 |
| | ChatCAD+ | 0.310 | 0.168 | 0.100 | 0.063 | **0.290** | 0.199 | 0.511 | 0.523 | **0.516** |
| | GPT-4V | 0.166 | 0.088 | 0.046 | 0.026 | 0.281 | 0.159 | 0.435 | **0.589** | 0.500 |
| | Med-Flamingo | 0.164 | 0.077 | 0.042 | 0.026 | 0.177 | 0.133 | 0.447 | 0.333 | 0.382 |
| | LLaVa-Med | 0.271 | 0.131 | 0.072 | 0.044 | 0.215 | 0.159 | 0.433 | 0.304 | 0.357 |
| | RadFM | 0.167 | 0.087 | 0.050 | 0.031 | 0.144 | 0.131 | 0.463 | 0.328 | 0.384 |
| | Flamingo* | 0.356 | 0.214 | 0.135 | 0.094 | 0.281 | 0.229 | 0.438 | 0.366 | 0.399 |
| | Ours | **0.383** | **0.231** | **0.147** | **0.098** | 0.287 | 0.242 | 0.511 | 0.493 | 0.502 |
| Medical Record | Cvt2DistilGPT2 | 0.303 | 0.189 | 0.127 | 0.091 | 0.261 | 0.248 | 0.551 | 0.437 | 0.487 |
| | ChatCAD+ | 0.179 | 0.090 | 0.050 | 0.031 | 0.227 | 0.123 | 0.456 | 0.588 | 0.513 |
| | GPT-4V | 0.093 | 0.051 | 0.028 | 0.016 | 0.234 | 0.103 | 0.420 | **0.654** | 0.512 |
| | Med-Flamingo | 0.164 | 0.078 | 0.044 | 0.027 | 0.169 | 0.127 | 0.516 | 0.478 | 0.496 |
| | LLaVa-Med | 0.238 | 0.116 | 0.063 | 0.039 | 0.218 | 0.149 | 0.499 | 0.410 | 0.450 |
| | RadFM | 0.118 | 0.050 | 0.026 | 0.015 | 0.079 | 0.085 | 0.278 | 0.113 | 0.161 |
| | Flamingo* | 0.397 | 0.265 | 0.178 | 0.129 | 0.327 | 0.217 | 0.562 | 0.469 | 0.511 |
| | Ours | **0.377** | **0.254** | **0.183** | **0.142** | **0.335** | **0.292** | **0.580** | 0.468 | **0.518** |
| Average | Cvt2DistilGPT2 | 0.265 | 0.164 | 0.109 | 0.078 | 0.237 | 0.230 | 0.545 | 0.367 | 0.461 |
| | ChatCAD+ | 0.388 | 0.284 | 0.228 | 0.193 | 0.387 | 0.315 | 0.543 | **0.632** | 0.577 |
| | GPT-4V | 0.245 | 0.178 | 0.138 | 0.113 | 0.374 | 0.266 | 0.529 | 0.608 | 0.579 |
| | Med-Flamingo | 0.174 | 0.097 | 0.065 | 0.048 | 0.185 | 0.145 | 0.400 | 0.330 | 0.353 |
| | LLaVa-Med | 0.226 | 0.127 | 0.083 | 0.060 | 0.210 | 0.173 | 0.436 | 0.318 | 0.361 |
| | RadFM | 0.106 | 0.054 | 0.031 | 0.020 | 0.100 | 0.096 | 0.316 | 0.178 | 0.225 |
| | Flamingo* | 0.472 | 0.350 | 0.286 | 0.248 | 0.422 | 0.370 | 0.572 | 0.500 | 0.532 |
| | Ours | **0.490** | **0.370** | **0.305** | **0.268** | **0.439** | **0.397** | **0.629** | 0.564 | **0.590** |

Table 6: Comparison of our model with other baselines on the test sets of *MIMIC-R3G*-test-B. B@n, M, R-L represent the NLG metrics BLEU, METEOR, and ROUGE-L respectively. P, R, F1 represent the CE metrics CheXpert precision, recall, and F1-score respectively. Flamingo* represents Flamingo model finetuned on training set of *MIMIC-R3G*.

## C.4 *DeMMo* ABLATION STUDY

We conduct ablation experiments to compare the performance of three other model designs. Table 5 reports the performance comparison. Under the same setting mentioned in section 6.1, we train and compare the performance of three other model designs. *DeMMo* outperforms the vanilla Flamingo without using medical vision encoders, showing the importance of adopting a medical vision encoder to enhance the domain-specific ability. The second baseline does not preserve the original Flamingo visual encoder like *DeMMo*, instead it directly replaces it with a medical vision encoder. The comparison results verify that preserving the original visual encoder can retain its general domain knowledge and hence help the performance. The third baseline trains the architecture design with both original Flamingo vision encoder and the medical vision encoder, but without any pathological guidance. Compared to this baseline, *DeMMo* achieves generally higher performance, which highlights the efficacy of the design of pathological guidance in enhancing the model's capabilities for medical domain-specific tasks.

## C.5 USE CASES

In this section we present inference results of out model on all proposed report generation tasks. Input text prompt to the model is the concatenation of context and instruction in arbitrary order.

---

### Inference Examples of No Context Generation

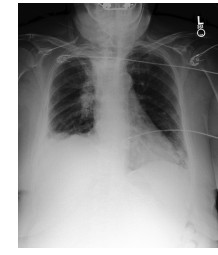

**Ground Truth Report:** (s50043351)
there is a right pleural effusion which is unchanged since prior exam. again seen is a right hilar opacity consistent with fibrosis, better assessed on recent ct. a subtle left lower lobe opacity is seen, which may represent atelectasis, but pneumonia cannot be excluded. the lungs are otherwise clear. the cardiomediastinal silhouette is unchanged from prior exam. visualized osseus structures are unremarkable.

- - - - - - - - - - - - - - - - - - - - - - - - - - - - - - - - - - - - - - - - -

**Instruction:**
Act as a radiologist and write a diagnostic radiology report for the patient based on their chest radiographs.

- - - - - - - - - - - - - - - - - - - - - - - - - - - - - - - - - - - - - - - - -

*DeMMo* **Output**
in comparison with study of ___, there is continued enlargement of the cardiac silhouette with evidence for elevated pulmonary venous pressure. opacification along the right lateral chest wall suggests increasing right pleural effusion with compressive atelectasis at the base. mild atelectatic changes are seen at the left base.

---

### Inference Examples of No Context Generation

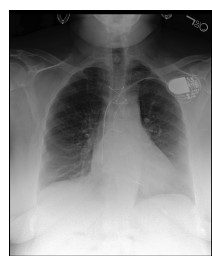

**Ground Truth Report:** (s56104633)
frontal and lateral views of the chest were obtained. patient is status post median sternotomy and cardiac valve replacement. dual lead left-sided pacemaker is seen with leads extending to the expected position of the right atrium and right ventricle. there may be minimal basilar atelectasis. no focal consolidation is seen. there is no pleural effusion or pneumothorax. the cardiac and mediastinal silhouettes are stable and unremarkable.

- - - - - - - - - - - - - - - - - - - - - - - - - - - - - - - - - - - - - - - - -

**Instruction:**
Based on the given chest X-ray images, please write a diagnostic radiology report for the patient.

- - - - - - - - - - - - - - - - - - - - - - - - - - - - - - - - - - - - - - - - -

*DeMMo* **Output**
pa and lateral views of the chest provided. left chest wall pacer device is again seen with pacer leads extending to the region of the right atrium and right

---

ventricle. midline sternotomy wires and mediastinal clips are again noted. the lungs are clear without focal consolidation, effusion or pneumothorax. the cardiomediastinal silhouette is normal. imaged osseous structures are intact. no free air below the right hemidiaphragm is seen.

## Inference Examples of Generation with Previous Visit as Context

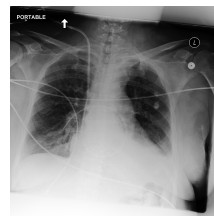

**Ground Truth Report:** (s59751598)
swan-ganz catheter has been advanced beyond the right hilum, and should be withdrawn for standard positioning, as discussed by telephone with dr. ___ at 9:45 a.m. on ___ new airspace opacity distal to the catheter tip could potentially represent pulmonary hemorrhage, but other etiologies such as atelectasis or aspiration are also possible. improving atelectasis in left lower lobe and persistent small left pleural effusion. incidental calcified granulomas within the left upper lobe.

- - - - - - - - - - - - - - - - - - - - - - - - - - - - - - - - - - - - - - - -

**Context:**
The patient has a previous visit with the report: endotracheal tube and nasogastric tube remain in standard position. swan-ganz catheter has been slightly withdrawn, with tip terminating in the central right hilar region. left sided catheter has been removed, with no visible pneumothorax. cardiac silhouette remains mildly enlarged, but previously reported mild edema has nearly resolved. bibasilar retrocardiac atelectasis is present with some improvement on the left, and small left pleural effusion is unchanged.

- - - - - - - - - - - - - - - - - - - - - - - - - - - - - - - - - - - - - - - -

**Instruction:**
Considering the patient's previous visit, please write a chest x-ray report for the patient based on the chest x-ray images:

- - - - - - - - - - - - - - - - - - - - - - - - - - - - - - - - - - - - - - - -

*DeMMo* **Output:**
in comparison with the study of ___, the endotracheal and nasogastric tubes have been removed. swan-ganz catheter remains in place. continued enlargement of the cardiac silhouette with indistinct pulmonary vessels suggesting elevated pulmonary venous pressure. opacification at the left base most likely reflects atelectasis and effusion, though in the appropriate clinical setting, supervening pneumonia would have to be considered. no gross evidence of pneumoperitoneum, though this is not an upright view.

## Inference Examples of Generation with Previous Visit as Context

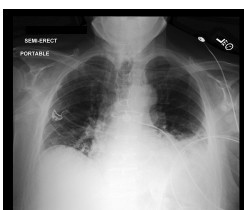

**Ground Truth Report:** (s58402174)
ap portable semi upright view of the chest. lung volumes are low limiting assessment. there is increased bibasilar atelectasis and bronchovascular crowding. overall cardiomediastinal silhouette is unchanged. the right upper extremity access picc line appears in unchanged position extending to the level of the cavoatrial junction. mild congestion is difficult to exclude in the correct clinical setting. no overt signs of edema.

- - - - - - - - - - - - - - - - - - - - - - - - - - - - - - - - - - - - - - - -

**Context:**
Medical report from the last visit: rounded bilateral mid lung opacities are again seen, grossly unchanged and likely reflect consolidative infectious process given history of septic emboli. there is unchanged bibasilar opacification, which is likely atelectasis with left greater than right effusions. cardiac silhouette is markedly enlarged, similar to the most recent prior. left picc terminates in the cavoatrial junction. median sternotomy wires are intact.

- - - - - - - - - - - - - - - - - - - - - - - - - - - - - - - - - - - - - - - -

**Instruction:**
Please write a diagnosis chest x-ray report by investigating the given chest x-ray images, referencing the patient's previous report:

- - - - - - - - - - - - - - - - - - - - - - - - - - - - - - - - - - - - - - - -

*DeMMo* **Output:**
portable semi-upright radiograph of the chest demonstrates low lung volumes which results in bronchovascular crowding. bibasilar atelectasis is unchanged. the cardiomediastinal and hilar contours are unchanged. there is no pneumothorax, pleural effusion, or consolidation. a left-sided picc line ends at the cavoatrial junction. median sternotomy wires are intact.

## Inference Examples of Generation with Patient's Medical Record as Context

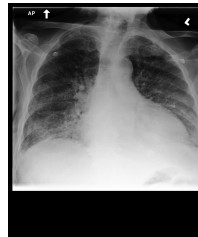

**Ground Truth Report:** (s54225810)
there is hilar congestion and diffuse bilateral ground glass opacities, most predominant at the bases, slightly improved from prior exam, and most consistent with pulmonary edema. an underlying pneumonia cannot be fully excluded. there are trace bilateral pleural effusions. there is no pneumothorax. the cardiac silhouette is moderately enlarged and unchanged from the prior exam. the mediastinal contours are normal.

**Context:**
The patient has the following medical history:
Medical History: 1. Multifocal Pneumonia 2. Pulmonary Edema 3. Heart Disease 4. Pleural Effusion Medical Tests: 1. Complete Blood Count (CBC): Elevated white blood cell count 2. Blood Culture: Possible bacterial growth indicating an infection 3. Sputum Culture: Presence of pathogenic bacteria indicating pneumonia 4. B-type Natriuretic Peptide (BNP) Test: Elevated BNP levels indicating heart failure

**Instruction:**
Examine the given chest x-ray images and patient's medical record and write a medical report detailing the findings.

*DeMMo* **Output:**
as compared to the previous radiograph, the patient has developed mild-to-moderate pulmonary edema. in addition, bilateral pleural effusions have newly occurred. the heart continues to be slightly enlarged. no evidence of pneumonia. no pneumothorax.

## Inference Examples of Generation with Patient's Medical Record as Context

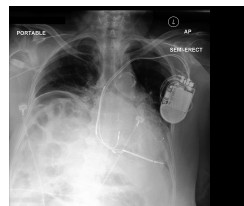

**Ground Truth Report:** (s58585557)
portable semi-upright radiograph of the chest demonstrates low lung volumes with resultant bronchovascular crowding. bibasilar consolidations may represent atelectasis or pneumonia in the appropriate clinical setting. the cardiomediastinal and hilar contours are unchanged. there is a new lucency beneath the right hemidiaphragm concerning for intra-abdominal free air. right-sided picc line and to the mid svc. unchanged position of the aicd. no pneumothorax.

**Context:**
The patient has following information: acute respiratory distress syndrome (ards), chronic obstructive pulmonary disease (copd)

**Instruction:**
Review the attached chest x-ray images and relevant patient information to write a detailed medical report:

*DeMMo* **Output:**
portable semi-upright radiograph of the chest demonstrates low lung volumes with resulting bronchovascular crowding. there is an area of increased opacification at the right lung base, which may represent atelectasis, however aspiration or pneumonia could be considered in the appropriate clinical setting. there is a probable small right-sided pleural effusion. the cardiomediastinal and hilar contours are unchanged. a right-sided internal jugular central venous line ends at the cavoatrial junction. a dual-chamber pacemaker is present, with leads in unchanged position. there is no pneumothorax. free air is seen under the right hemidiaphragm, consistent with peritoneal dialysis.

## Inference Examples of Generation with Template as Context

**Ground Truth Report:** (s52030252)

lungs:
- parenchyma: clear
- nodules/masses: absent
cardiomediastinal contour:
- heart size: enlarged
- mediastinal shape: normal
pleura:
- pleural effusion: absent
- pleural thickening: absent
bones:
- ribs/spine/clavicles/scapulae: normal
diaphragm and abdomen:
- diaphragm: normal contour
- abdominal component: not visible
soft tissues and other observations:
- soft tissue: normal
- foreign bodies: absent
- additional findings: atelectatic changes are seen at both lung bases, right more than left. there are improved lung volumes, notably on the left, potentially reflecting increased ventilatory pressure. no new parenchymal opacities. unchanged moderate cardiomegaly with tortuosity of the thoracic aorta.

---

**Context:**
Template:
Lungs:
- Parenchyma: [Clear — Consolidation — Interstitial markings — Other: please specify]
- Nodules/Masses: [Absent — Present]
If present, complete the following:
- Number of Nodules/Masses: [#]
- Size of the largest Nodule/Mass: [# mm — # cm]
- Location: [Right Upper Lobe — Right Middle Lobe — Right Lower Lobe — Left Upper Lobe — Left Lower Lobe; specify segment if known]
- Characteristics: [Well-defined — Spiculated — Cavitary — Calcified — Other: [specify]]
- Density: [Solid — Ground glass — Mixed — Other: [specify]]
Cardiomediastinal Contour:
- Heart Size: [Normal — Enlarged]
- Mediastinal Shape: [Normal — Widened — Other: please specify]
Pleura:
- Pleural Effusion: [Absent — Present]
If present, specify side and approximate volume if possible
- Pleural Thickening: [Absent — Present]
If present, describe extent and location
Bones:
- Ribs/Spine/Clavicles/Scapulae: [Normal — Fracture(s) — Lesion(s) — Other: [specify]]
Diaphragm and Abdomen:
- Diaphragm: [Normal contour — Elevated hemidiaphragm — Other: please specify]
- Abdominal Component: [Not visible — Gas under diaphragm — Other: please specify]
Soft Tissues and Other Observations:
- Soft Tissue: [Normal — Abnormality noted: please specify]
- Foreign Bodies: [Absent — Present: please specify location and appearance]
- Additional Findings: [None — Specify: please specify]
Support Devices: [None — pacemaker — PICC — tube — catheter — other]

---

**Instruction:**
Please fill this chest x-ray diagnostic report template based on the give chest x-ray radiographs.

---

*DeMMo* **Output:**
lungs:
- parenchyma: interstitial markings
- nodules/masses: absent
cardiomediastinal contour:
- heart size: enlarged
- mediastinal shape: normal
pleura:
- pleural effusion: absent
- pleural thickening: absent
bones:
- ribs/spine/clavicles/scapulae: normal
diaphragm and abdomen:
- diaphragm: normal contour
- abdominal component: not visible
soft tissues and other observations:
- soft tissue: abnormality noted: patchy opacities in the lung bases
- foreign bodies: absent
- additional findings: low lung volumes
support devices: tracheostomy tube, right internal jugular central venous catheter

## Inference Examples of Generation with Template as Context

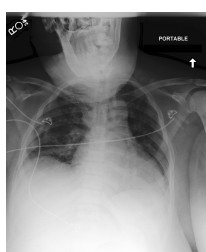

**Ground Truth Report:** (s55646831)
Bronchovascular markings: Right upper lobe scarring/chronic fibrosis in the right greater than left upper lobes.
Rest of the visualised lung fields: There are low lung volumes. New opacity in the left mid-to-lower lung.
Bilateral hilum:
Cardiac silhouette: The cardiac and mediastinal silhouettes are stable.
Costophrenic angles: No pleural effusion is seen.
Visualised bones & soft tissues:
Support Devices:

**Context:**
Template:
Bronchovascular markings:
Rest of the visualised lung fields:
Bilateral hilum:
Cardiac silhouette:
Costophrenic angles:
Visualised bones & soft tissues:
Support Devices:

**Instruction:**
Write a diagnostic radiology report for the patient based on their chest radiographs following the given template:

*DeMMo* **Output:**
bronchovascular markings: diffuse airspace consolidation in the right lung
rest of the visualized lung fields: low volume, no evidence of cardiomegaly or pleural effusion
bilateral hilum: no large effusion or pneumothorax
cardiac silhouette: appears normal
costophrenic angles: clear
visualized bones & soft tissues: intact
support devices: none

## Inference Examples of Report Revision

**Ground Truth Report:** (s57996680)
low lung volumes are present. this accentuates the size of the cardiac silhouette which is likely mildly enlarged. mediastinal and hilar contours are likely within normal limits. a right brachiocephalic venous stent is re- demonstrated. there is crowding of the bronchovascular structures with probable mild pulmonary vascular congestion. no pleural effusion or pneumothorax is identified.

**Context:**
Incorrect report: Normal lung volumes are present. The cardiac silhouette is of regular size. Mediastinal and hilar contours are likely mildly enlarged. No evidence of a right brachiocephalic venous stent. There is no crowding of the bronchovascular structures and no signs of pulmonary vascular congestion. Pleural effusion and pneumothorax are identified.

**Instruction:**
1. The lung volumes are described as normal in the incorrect report while they are low in the correct report. This needs to be fixed.
2. The incorrect report incorrectly states that there is no evidence of a right brachiocephalic venous stent, while the correct report states that it is re-demonstrated.
3. The incorrect report mistakenly identifies pleural effusion and pneumothorax, which are not present as per the correct report.
4. The incorrect report states that there is no crowding of the bronchovascular structures and no signs of pulmonary vascular congestion, which is contrary to the correct report. This needs correction.

*DeMMo* **Output:**
low lung volumes are present. this accentuates the size of the cardiac silhouette which is enlarged. mediastinal and hilar contours are likely within normal limits. a right brachiocephalic venous stent is re- demonstrated. there is crowding of the bronchovascular structures with mild pulmonary vascular congestion. no pleural effusion or pneumothorax is identified.

---

**Inference Examples of Report Revision**

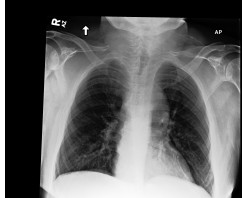

**Ground Truth Report:** (s56541072)

heart size is normal. the aorta is tortuous. unchanged widening of the mediastinum attributable to mediastinal lipomatosis is re- demonstrated. hilar contours are unremarkable. pulmonary vasculature is not engorged. lungs are clear. no pleural effusion, focal consolidation or pneumothorax is demonstrated. there are no acute osseous abnormalities.

- - - - - - - - - - - - - - - - - - - - - - - - - - - - - - - - - - - - - - - - - - - - - - - - - - - - - -

**Context:**

Report: Heart size is enlarged. The aorta is not tortuous. No evidence of mediastinal lipomatosis. Hilar contours are abnormal. Pulmonary vasculature is engorged. Lungs have a small focal consolidation in the right lower lobe. Pleural effusion is present. There are multiple acute osseous abnormalities.

- - - - - - - - - - - - - - - - - - - - - - - - - - - - - - - - - - - - - - - - - - - - - - - - - - - - - -

**Instruction:**

1. Revise the findings regarding heart size, aorta, mediastinum, hilar contours, pulmonary vasculature and lungs condition. The incorrect report inaccurately states these as abnormal when they are actually normal.
2. Correct the diagnosis of the presence of a small focal consolidation in the right lower lobe, pleural effusion, and multiple acute osseous abnormalities. These conditions are not present according to the correct report.
3. Include the detail about the mediastinal lipomatosis, as it is important information that was omitted in the incorrect report.

- - - - - - - - - - - - - - - - - - - - - - - - - - - - - - - - - - - - - - - - - - - - - - - - - - - - - -

*DeMMo* **Output:**

heart size is normal. the aorta is mildly tortuous. widening of the right paratracheal stripe is unchanged, and could suggest underlying lymphadenopathy or mediastinal fat or lipomatosis. hilar contours are normal. pulmonary vasculature is not engorged. lungs are clear without focal consolidation. no pleural effusion or pneumothorax is present. there are no acute osseous abnormalities. heart size is normal. the aorta is tortuous. unchanged widening of the mediastinum attributable to mediastinal lipomatosis is re- demonstrated. hilar contours are unremarkable. pulmonary vasculature is not engorged. lungs are clear without focal consolidation. no pleural effusion or pneumothorax is present. there are no acute osseous abnormalities.

---

# D   COLLECTED REPORT TEMPLATES

We collected 10 templates with the help of medical professionals. Six of them are real-world templates sourced from published papers, RSNA's recommended RadReport Templates, and web searches. Additionally, we generated 4 templates using GPT. Similar to the data annotation process, our templates, whether generated or sourced from real-world data, are revised and validated by licensed medical professionals. This process adheres to the IHE Radiology Technical Framework Supplement's guidelines (IHE, 2018), encompassing text fields, number fields, and choice fields. The templates used are shown below. Templates without reference are generated by GPT and revised by medical professionals.

---

**Template 1 (Gunn et al., 2015)**

REPORT: [Imaging Protocol]
COMPARISON: [None/Compare with former image]
FINDINGS:
    Lines/tubes:
    Lungs:
    Pleura:
    Heart and mediastinum:
    Bones:
    Support Devices:

---

**Template 2 (DrLogy, 2024)**

Findings:
    Bronchovascular markings:
    Rest of the visualised lung fields:
    Bilateral hilum:
    Cardiac silhouette:
    Costophrenic angles:
    Visualised bones & soft tissues:
    Support Devices:

**Template 3 (CP, 2011)**

Comparison:
  - [ ]None.
  - [ ]Compare to historical Report.

Findings:
  Lungs:
  - [ ]The lungs are clear.
  - [ ]The inspiratory volumes are small, and this probably accounts for some vascular crowding and atelectasis at the bases.
  - [ ]There is focal opacity at the right lung base most likely representing right lower lobe atelectasis.
  - [ ]There is focal opacity at the right lung base most likely representing a combination of a moderate right pleural effusion and associated passive right lower lobe atelectasis.
  - [ ]There is a focal opacity at the left lung base most like representing left lower lobe atelectasis.
  - [ ]There is a focal opacity at the left lung base, characteristic of a combination of a moderate left pleural effusion and associated atelectasis.
  - [ ]There is patchy opacity at both lung bases characteristic of atelectasis.
  - [ ]There is patchy opacity at both lung bases characteristic of a combination of atelectasis and effusions.
  - [ ]There is vascular congestion with increased interstitial markings findings indicating mild cardiogenic edema.
  - [ ]There is vascular congestion with mixed interstitial and patchy alveolar opacities, findings indicating moderate cardiogenic edema.
  - [ ]There is extensive alveolar consolidation in the lungs bilaterally, most likely representing pulmonary edema. This is probably on the basis of severe congestive heart failure but could be a result of noncardiogenic causes.
  - [ ]There are multiple patchy areas of consolidation, widely scattered about the lungs bilaterally. This most likely represents a multifocal pneumonia.
  - [ ]There is patchy opacity at both lung bases characteristic of a combination of atelectasis and effusions. There is mild vascular and interstitial prominence, likely reflecting mild pulmonary edema.
  - [ ]There is mild pulmonary vascular engorgement without pulmonary edema.

  Heart:
  - [ ]The heart is normal in size.
  - [ ]There is mild cardiomegaly.
  - [ ]There is moderate cardiomegaly.
  - [ ]There is severe cardiomegaly.
  - [ ]There is marked cardiomegaly.
  - [ ]The heart is top normal in size.

  Mediastinum:
  - [ ]The mediastinum is within normal limits.
  - [ ]Atherosclerotic calcifications are seen in the aorta.
  - [ ]The aorta appears tortuous, a finding usually associated with either atherosclerosis or systemic hypertension.
  - [ ]The aortic contour is quite prominent, a finding likely indicating either an aortic aneurysm or dissection.
  - [ ]Post-operative changes are present in the mediastinum.
  - [ ]Degenerative changes are present in the thoracic spine.

  Support Devices:
  - [ ]None
  - [ ]pacemaker
  - [ ]PICC
  - [ ]tube
  - [ ]catheter
  - [ ]other

**Template 4 (Schmidt, 2017)**

Comparison:
  Comparison Study:
  - [ ]None.
  - [ ]Compare to historical Report.

Findings:
  Lungs:
  -[ ] The lungs are clear.
  -[ ] Subsegmental atelectasis is present at both bases.
  -[ ] Bibasilar opacities represent small bilateral pleural effusions with overlying atelectasis.
  -[ ] Mild pulmonary vascular congestion is present. There is no evidence of associated pulmonary edema.
  -[ ] Mild diffuse interstitial pulmonary edema is present, likely cardiogenic.
  -[ ] Moderate alveolar pulmonary edema is present, likely cardiogenic.
  -[ ] Marked diffuse pulmonary edema and consolidation are present.
  -[ ] Subsegmental atelectasis is present at the left base.
  -[ ] Subsegmental atelectasis is present at the right base.
  -[ ] An opacity at the left base represents a small pleural effusion with overlying atelectasis.
  -[ ] An opacity at the right base represents a small pleural effusion with overlying atelectasis.
  -[ ] Small bilateral pleural effusions are present with overlying atelectasis. Mild cardiogenic interstitial edema also is present.
  -[ ] The inspiratory volumes are small, which probably explains increased interstitial opacity and atelectasis at the bases.
  -[ ] Other.

  Pleural Spaces:
  -[ ] No pleural abnormalities are listed.

-[ ] Trace bilateral pleural effusions are present.
-[ ] Small bilateral pleural effusions are present.
-[ ] Moderate bilateral pleural effusions are present.
-[ ] Large bilateral pleural effusions are present.
-[ ] Other.

Heart:
-[ ] The heart is normal in size.
-[ ] The heart is mildly enlarged.
-[ ] The heart is moderately enlarged.
-[ ] The heart is markedly enlarged.
-[ ] Other.

Mediastinum:
-[ ] The mediastinal contours are normal.
-[ ] The thoracic aorta is tortuous.
-[ ] Calcifications are present in the thoracic aorta.
-[ ] The thoracic aorta is tortuous and calcified.
-[ ] Other.

Osseours Structures:
-[ ] There are no osseous abnormalities.
-[ ] Degenerative changes are present in the thoracic spine.
-[ ] A mild thoracic levoscoliosis is present.
-[ ] A mild thoracic dextroscoliosis is present.
-[ ] A mild S-shaped thoracolumbar scoliosis is present.
-[ ] Other.

Additional Findings:
-[ ] None.
-[ ] Additional Findings:

Support Devices:
- [ ]None
- [ ]pacemaker
- [ ]PICC
- [ ]tube
- [ ]catheter
- [ ]other

## Template 5 (Mityul et al., 2018)

Modality: X-rays
Part: Chest

Findings:
Bony Cage: [Normal/Other findings]
Soft tissue of Chest: [Normal/Other findings]
Trachea: [In Midline/Other findings]
Lungs: [Both Lung fields are equally translucent/Other findings]
Heart: [Cardiac size and contour are normal/Other findings]
Hilum & Mediastinum: [Normal/Other findings]
Costphrenic and Cardiophrenic angles: [Clear/Other findings]
Support Devices: [None/Findings]
Other: [Nil/Other findings]

## Template 6 (Marcovici & Taylor, 2014)

COMPARISON: [None./Comparison]
FINDINGS:
Lungs/pleura: [Normal./Other findings]
Heart/mediastinum: [Normal./Other findings]
Bones/Soft tissues: [Normal./Other findings]
Support Devices: [None./Other findings]

## Template 7

Findings:
Lungs:
Parenchyma: [Clear | Infiltrates | Consolidation | Nodules]
Pleura: [Normal | Thickening | Effusion]
Interstitial Markings: [Normal | Increased]

Heart:
Size: [Normal | Enlarged]
Contours: [Normal | Abnormal]

```
Mediastinum:
    Width: [Normal | Wide]
    Contour: [Normal | Abnormal]

Bones:
    Ribs: [Normal | Fracture | Lesion]
    Spine: [Normal | Degenerative changes | Fracture | Lesion]
    Clavicles: [Normal | Fracture | Lesion]

Diaphragm:
    Position: [Normal | Elevated]
    Contour: [Normal | Abnormal]

Soft Tissues: [Normal | Abnormal]

Support Devices: [None | pacemaker | PICC | tube | catheter | other]
```

## Template 8

```
Findings:
    Heart: [Normal size and contour | Enlarged | Other]
    Mediastinum: [Normal contour | Widened | Mass | Other]
    Lungs:
        - Parenchyma: [Clear | Consolidation | Interstitial markings | Nodule(s) | Mass | Other]
        - Effusion: [Absent | Small | Moderate | Large]
            if Effusion is not Absent:
                - Location: [Right | Left | Bilateral]
                - Estimated volume: [<=100 mL | 101-500 mL | 501-1000 mL | >1000 mL]
        - Pneumothorax: [Absent | Present]
                - Size: [<# cm at apex | # cm]
    Bones: [Normal | Fracture(s) | Lytic lesions | Other abnormalities]
    Soft Tissues: [Normal | Swelling | Mass | Air | Other abnormalities]
    Diaphragm: [Well-defined | Elevated | Blurred | Irregular | Other]
    Pleura: [Normal | Thickening | Plaque | Calcification | Other]
    Support Devices: [None | pacemaker | PICC | tube | catheter | other]
    Other findings: [Provide details if any other abnormalities are noted]
```

## Template 9

```
Findings:

    Heart:
        - Size: [Normal | Enlarged]
        - Contour: [Normal | Abnormal]

    Lungs:
        - Lung Fields: [Clear | Consolidation | Infiltrates | Pleural Effusion]
        - Nodules/Masses: [None | Single | Multiple]
            - If applicable, provide details:
                - Location: [Right Upper Lobe; Right Middle Lobe; Right Lower Lobe; Left Upper Lobe; Left Lower Lobe]
                - Size: [# cm]
                - Characteristics: [Smooth; Spiculated; Calcified]

    Pleura:
        - Pleural Lines: [Normal | Thickened]
        - Pleural Effusion: [None | Right | Left | Bilateral]

    Mediastinum:
        - Mediastinal Width: [Normal | Enlarged]
        - Mediastinal Masses: [No | Yes]
            - If applicable, provide details:
                - Location: [Anterior; Middle; Posterior]
                - Size: [# cm]
                - Characteristics: [Smooth; Irregular]

    Bones and Soft Tissues:
        - Ribs: [Normal | Fracture | Lesions]
        - Spine: [Normal | Degenerative Changes | Fracture | Lesions]
        - Soft Tissues: [Normal | Abnormal]

    Support Devices: [None | pacemaker | PICC | tube | catheter | other]
```

**Template 10**

Findings:
    Lungs:
      - Parenchyma: [Clear | Consolidation | Interstitial markings | Other: please specify]
     - Nodules/Masses: [Absent | Present]
       {If present, complete the following:}
        - Number of Nodules/Masses: [#]
        - Size of the largest Nodule/Mass: [# mm | # cm]
        - Location: [Right Upper Lobe | Right Middle Lobe | Right Lower Lobe | Left Upper Lobe | Left Lower Lobe; specify segment if known]
        - Characteristics: [Well-defined | Spiculated | Cavitary | Calcified | Other: [specify]]
        - Density: [Solid | Ground glass | Mixed | Other: [specify]]

    Cardiomediastinal Contour:
     - Heart Size: [Normal | Enlarged]
     - Mediastinal Shape: [Normal | Widened | Other: please specify]

    Pleura:
     - Pleural Effusion: [Absent | Present]
      {If present, specify side and approximate volume if possible}
     - Pleural Thickening: [Absent | Present]
      {If present, describe extent and location}

    Bones:
     - Ribs/Spine/Clavicles/Scapulae: [Normal | Fracture(s) | Lesion(s)| Other: [specify]]

    Diaphragm and Abdomen:
     - Diaphragm: [Normal contour | Elevated hemidiaphragm | Other: please specify]
     - Abdominal Component: [Not visible | Gas under diaphragm | Other: please specify]

    Soft Tissues and Other Observations:
     - Soft Tissue: [Normal | Abnormality noted: please specify]
     - Foreign Bodies: [Absent | Present: please specify location and appearance]
     - Additional Findings: [None | Specify: please specify]

    Support Devices: [None | pacemaker | PICC | tube | catheter | other]

## E   MANUALLY DESIGNED INSTRUCTIONS

Here we show the manually designed instruction we used in the *MIMIC-R3G*

**Instructions for No Context Report Generation**

- Act as a radiologist and write a diagnostic radiology report for the patient based on their chest radiographs
- Generate a comprehensive radiology report based on the chest X-ray images, detailing any findings and observations.
- Using the chest X-ray images provided, write a complete radiology report.
- Create a diagnostic report for the patient based on their chest radiographs.

**Instructions for Report Revision**

- Revise the medical report based on the chest x-ray radiographs and these instructions: {instructions}
- Fix this incorrect medical report of these chest x-ray images using these guidelines: {instructions}
- Update the medical report of the given chest x-ray images with these changes: {instructions}
- Based on the given chest x-ray images, edit this medical report following these suggestions: {instructions}
- Apply these revisions to the given medical report of the chest x-ray radiographs: {instructions}
- Refine the given medical report of the chest x-ray images with these improvements: {instructions}
- Enhance the medical report by incorporating these notes: {instructions}
- Revise the medical report based on the chest x-ray radiographs considering these recommendations: {instructions}
- Fix the given incorrect medical report based on the chest x-ray images and these instructions: {instructions}

**Instructions for Templated Report Generation**

- Please act as a radiologist and write a diagnostic radiology report for the patient based on their chest radiographs, the format should follow the template:{template}
- Write a diagnostic radiology report for the patient based on their chest radiographs following the given template:{template}
- Please fill the following chest x-ray radiology template based on the given chest x-ray images:{template}
- Template:{template} Please fill this chest x-ray diagnostic report template based on the give chest x-ray radiographs.

- Template: {template} Given this template, please fill it after investigating the given chest x-ray radiology report.
- Referencing the given chest x-ray images, please fill the following chest x-ray report template:{template}

### Instructions for Previous Radiology Image and Report as Context

- Previous medical report:{previous_report} Act as a radiologist and write a diagnostic radiology report for the patient based on their chest radiographs and previous medical report:
- Medical report from the last visit:{previous_report} Please write a diagnostic radiology report for the patient based on their chest radiographs considering the report from last visit:
- The patient has a previous visit with the report:{previous_report} Considering the patient's previous report, please write a chest x-ray report for the patient based on the chest x-ray images:
- Act as a radiologist and write a diagnosis chest x-ray report by inspecting patient's chest x-ray images and previous report. The patient's previous report:{previous_report}
- Please write a diagnosis chest x-ray report by investigating the given chest x-ray images, referencing the patient's previous report:{previous_report}

### Instructions for Medical Records and Lab Tests as Context

- The patient has the following medical conditions and exam result: {history} Examine the given chest x-ray images and patient's medical conditions, and write a medical report detailing the findings:
- The patient has following information: {history} Review the attached chest x-ray images and relevant patient information to write a detailed medical report:
- Medical conditions of the patient: {history} Based on the chest x-ray images and patient's medical details, draft a detailed diagnostic medical report:
- Given that the patient has the following medical history: {history}, write a detailed medical report for the patient based on the given medical history and chest x-ray radiographs:
- The patient has the following medical record: {history}, combine with the chest x-ray images, write a detailed diagnostic medical report for the patient:

## F    LIMITATIONS AND FUTURE WORKS

For the report revision task, our pipeline generates modifications in reverse from ground truth reports. Although human validation indicates a 97% acceptance rate, we cannot guarantee that the generated modifications accurately reflect the distribution of real-world errors made by human or AI report generation systems. Future work could focus on recording real-world clinical procedures where human radiologists revise reports generated by AI systems or written by junior radiologists, to better capture the nature of these errors.

For the task of using medical records and lab tests as context, although the MIMIC-IV dataset provides EHR data for patients in MIMIC-CXR, we opted to generate synthetic medical records primarily due to the significant effort required to backtrack and match the corresponding MIMIC-CXR studies with their associated hospital stays in MIMIC-IV. Methods such as [2] attempt to approximate the correspondence between CXR and EHR data; however, without a direct identification ID, the accuracy of these methods remains uncertain. Even with such linking methods, 55.99% of MIMIC-CXR studies could not be matched to a specific stay in the MIMIC-IV dataset. While our generated context achieved a 99.5% acceptance rate in human validation, it is important to note that the distribution of generated data may not perfectly reflect the true distribution. Future work could focus on reorganizing MIMIC-CXR and MIMIC-IV so that all EHR data in MIMIC-IV can be utilized in MIMIC-CXR, or similarly, on collecting and building datasets with available EHR data for report generation and other related tasks.

For *DeMMo*, as introduced in previous sections, our method is a pure generation method without encompassing extra generation priors such as labels from a classifier. In contrast, methods such as Zhao et al. (2023) and You et al. (2021) utilize an image classifier to extract disease labels prior to generation, which ensures diagnostic correctness. Tanida et al. (2023) leverages object detector and use the extracted abnormal regions to guide generation, which also shows promising result. This presents a limitation, as our model's diagnostic accuracy may not be as reliable as methods employing guidance from high accuracy classifiers. Therefore, future works may focus on fusing the model with extra generation prior or guidance to further improve clinical efficacy.

We also observe that our *DeMMo* approach can be generalized to other domains as well using other domain-specific vision encoders. A potential future direction could entail utilizing a CT scan en-

coder for CT report generation, or developing a universal medical vision encoder for a more unified medical report generation tasks.

## G ETHICS AND GPT DETAILS

The *MIMIC-R3G* dataset is derived from the MIMIC-CXR dataset, which was approved by the Institutional Review Board (IRB) of Beth Israel Deaconess Medical Center (BIDMC), Boston, MA. Following the training and guidelines provided by MIMIC, this project is classified as secondary research based on de-identified MIMIC data. Since the purpose of annotation is strictly for data quality control and not related to understanding user behaviors, characteristics, or preferences, the annotation process is not subject to additional IRB review. To comply with the MIMIC-CXR Data Use Agreement (DUA) and PhysioNet guidelines, all data generation processes were conducted using a HIPAA-compliant Azure OpenAI Service without human review. We use the chat completions API with gpt-4-32k as the underlying engine hosted on Azure OpenAI Service. All researchers and human annotators involved in the research have signed the DUA for MIMIC-CXR and have been approved to access the data. Authors of this project bear all responsibility in case of violation of rights.

