# OpenReview forum: "Benchmark Dataset for Radiology Report Generation with Instructions and Contexts"
_ICLR.cc/2025/Conference — Submitted to ICLR 2025_

### Official Review · Reviewer_e4Ay · 2024-10-30

**Soundness:** 2
**Presentation:** 3
**Contribution:** 3
**Rating:** 5
**Confidence:** 4

**Summary:**

This paper defines a new task, instructional report generation, based on real-world clinical needs and encompassing five subtasks. Leveraging the open-source MIMIC-CXR dataset, the authors designed an automated data generation method to construct a dedicated dataset and benchmark (MIMIC-R3G) for this task. The medical multimodal model, DeMMo, was trained on this dataset and demonstrated superior performance in benchmark comparisons with existing models.

**Strengths:**

+ The paper demonstrates high originality and academic importance by introducing the instructional report generation task with an innovative design.
+ The task is divided into five clinically meaningful subtasks, providing practical guidance for future research on report generation, and inspiring directions that align more closely with clinical practice.
+ The paper is well-organized, with a clear sequence of task, dataset/benchmark, and model, offering a coherent and complete narrative that aids reader comprehension.

**Weaknesses:**

+ **Insufficient Validation of Clinical Relevance for Subtasks**: While the paper emphasizes the dataset and benchmark, it lacks fair experiments to validate the practical significance of each subtask. For instance, when comparing different contexts (such as without context, previous visit, medical records, and lab tests as context), it would be beneficial to compare the performance on the same samples across subtasks to demonstrate the actual clinical relevance of these contexts.
+ **Lack of Task-Specific Model Adaptation**: As a dataset-focused paper, the model design lacks task-specific adaptation, which might detract from the paper's primary focus.
+ **Other Issues**: The symbol $I_i$ in line 191 should be $C_i$, and the numeric values in line 406 are inconsistent; $3846$ does not equal $2246 + 600$.

**Questions:**

In addition to the issues mentioned in the Weaknesses section, there are also the following concerns:
+ What is the clinical significance of report revision? Are there examples to illustrate this?

---

> ### Author Response · Authors · 2024-11-27
>
> Thank you for your constructive review! Please see our response below:
>
> # Clinical Relevance
> > Insufficient Validation of Clinical Relevance for Subtasks: While the paper emphasizes the dataset and benchmark, it lacks fair experiments to validate the practical significance of each subtask. For instance, when comparing different contexts (such as without context, previous visit, medical records, and lab tests as context), it would be beneficial to compare the performance on the same samples across subtasks to demonstrate the actual clinical relevance of these contexts.
>
> While we use the same samples to generate context for different tasks, the tasks themselves are designed to be relatively independent and orthogonal, focusing on distinct clinical scenarios. Our aim is not to enhance model performance on one task by leveraging data from another but rather to evaluate models' abilities in diverse and independent contexts. Even if there were performance variations across tasks using shared data, such differences would not necessarily provide direct insights into clinical relevance, as they could stem from task-specific complexities or model biases rather than real-world applicability. We appreciate the reviewer’s point and will consider incorporating additional analyses to better illustrate the practical significance of these contexts in future work.
>
>
> # Task Specific Model
> > Lack of Task-Specific Model Adaptation: As a dataset-focused paper, the model design lacks task-specific adaptation, which might detract from the paper's primary focus.
>
> Our model is specifically designed to address the unified task of report generation with contextual text input, which aligns with the primary focus of our proposed dataset. While it is not heavily tailored for individual subtasks, this generality ensures a consistent evaluation framework across the diverse tasks in our dataset, emphasizing the versatility and robustness of the model. We believe that this unified approach provides a solid foundation for benchmarking and encourages further research into task-specific adaptations, which can be explored in future work.
>
>
>
>
>
>
> # Typos
> > Other Issues: The symbol $I_i$ in line 191 should be $C_i$, and the numeric values in line 406 are inconsistent; $3846$ does not equal $2246 + 600$.
>
> Thanks for pointing these out. We have updated our paper accordingly.
>
>
>
> # Significance of Revision Task
> > What is the clinical significance of report revision? Are there examples to illustrate this?
>
> Currently, AI report generation systems are far from perfect, with many methods achieving only about a 0.5 F1 score, as shown in Table 2 (No Context section) of our main paper. This shows the necessity of having a human in the loop. However, manually revising the entire generated report can be exhausting, therefore this task is designed to reduce effort by allowing humans to provide brief and simple instructions for AI to follow and do the actual revision.
> Therefore there can be two AI systems operate under entirely different conditions, and the human revision suggestions provided to the second AI give it a distinct advantage over the first. If we trust in the correctness of human knowledge and the provided instruction, the revised report in this setting, while perhaps not perfect, should be more accurate than the output of the first AI.
> We believe that while the first AI primarily focuses on providing a general diagnosis for the entire image, the secondary AI specializes in following instructions at various levels. These range from simple text modifications to maintaining report consistency and even re-diagnosing specific locations or abnormalities based on human instructions. If we trust in the correctness of human knowledge and the provided instruction, the revised report in this setting, while perhaps not perfect, should be more accurate than the output of the first AI.
>
> For example, an example conversation between a radiologists and a report revision model could be as follow:
> > **Radiologist:**
> > (chest x-ray images)
> >
> > **First AI Response:**
> > portable ap upright chest radiograph obtained. there has been interval removal of a right ij central venous catheter with its tip located in the distal svc or cavoatrial junction. no pneumothorax identified. otherwise, no change.
> >
> > **Radiologist:**
> > sternotomy wires and mediastinal clips are present, reflect the placement of the ij cvc.
> >
> > **Second AI Response:**
> > portable ap upright chest radiograph obtained. midline sternotomy wires and mediastinal clips are again noted. there has been interval placement of a right ij central venous catheter with its tip located in the distal svc or cavoatrial junction. no pneumothorax. otherwise, no change.

---

> > ### Comment · Reviewer_e4Ay · 2024-11-27
> >
> > Thank you for your response! After reviewing your reply, I still believe that not conducting experiments to explore the correlations between report generation tasks under different contexts remains a weakness of this paper. Therefore, I will not change my assessment.

---

### Official Review · Reviewer_tNne · 2024-11-03

**Soundness:** 2
**Presentation:** 2
**Contribution:** 2
**Rating:** 3
**Confidence:** 4

**Summary:**

This paper extends the radiology report dataset MIMIC-CXR to a new benchmark dataset called MIMIC-R3G which adds instructions and context, in addition to images and radiology reports. The aim of MIMIC-R3G is to benchmark the capabilities of models on radiology report writing, revising, template-based writing, and writing with context information (previous radiology reports, medical records, lab tests), respectively. The paper also proposes a new method DeMMo which is evaluated on MIMIC-R3G and demonstrates good performance compared to baseline methods.

**Strengths:**

This paper presents an extension of the MIMIC-CXR dataset into more scenarios (simulating how clinicians may write radiology reports with instructions and context) which can help evaluate the capabilities of models in writing and revising radiology reports, from the technical perspective (but probably not from the clinical perspective, see Weakness).

A new method which demonstrates good performance on the new dataset MIMIC-R3G.

**Weaknesses:**

The paper's writing requires improvement, especially in terms of clarity. For instance, when the "pipeline" is mentioned on Page 4, it is neither defined nor illustrated, leaving readers uncertain about its components and workflow. Including an illustration or a detailed explanation of the pipeline would help understanding.

The motivation of the paper is also unclear. While it emphasizes supporting real-world clinical practice in Line 049 for radiology report generation (with instructions and context), the expectation is to collect a new dataset originating from actual clinical workflows where clinicians follow specific instructions to write radiology reports. However, the new dataset, MIMIC-R3G, is an extension of MIMIC-CXR rather than a newly collected dataset from real-world practices involving radiologists. Consequently, the setup of this work does not adequately justify its motivation and limits its overall contribution. It might be beneficial to collect a small-scale, new dataset from real-world practice and use MIMIC-R3G as a supplementary large-scale, silver-standard dataset.

Furthermore, the paper focuses on radiology report generation with a proposed method specifically designed and optimized for this purpose. It remains unclear how the pipeline used to produce this new dataset and the new method, DeMMo, can be applied to other domains, which would be of interest to a broader ICLR audience.

**Questions:**

What is "correction" in Line 264?

Is templated-based writing actually useful? Do radiologists write reports following templates? Figure 1 also shows a relatively low acceptance rate.

Table 2 seems to demonstrate that using previous radiology reports and medical records is not helpful (compared to no context). Any insights?

---

> ### Author Response · Authors · 2024-11-27
>
> Thank you for the constructive review! Please see our response below:
>
> # Clarity
> > The paper's writing requires improvement, especially in terms of clarity. For instance, when the "pipeline" is mentioned on Page 4, it is neither defined nor illustrated, leaving readers uncertain about its components and workflow. Including an illustration or a detailed explanation of the pipeline would help understanding.
>
> We appreciate the feedback. Section 4.2 provides a detailed explanation of our data generation process, including examples of system messages, prompts, and outputs, with additional examples in Appendix B. We believe this adequately covers the workflow of our data generation pipeline. However, we will consider providing further clarification if needed to ensure better understanding.
>
>
> # Motivation
> > The motivation of the paper is also unclear. While it emphasizes supporting real-world clinical practice in Line 049 for radiology report generation (with instructions and context), the expectation is to collect a new dataset originating from actual clinical workflows where clinicians follow specific instructions to write radiology reports. However, the new dataset, MIMIC-R3G, is an extension of MIMIC-CXR rather than a newly collected dataset from real-world practices involving radiologists. Consequently, the setup of this work does not adequately justify its motivation and limits its overall contribution. It might be beneficial to collect a small-scale, new dataset from real-world practice and use MIMIC-R3G as a supplementary large-scale, silver-standard dataset.
>
> 1. **Collecting real-world clinical distributions is extremely challenging**: We recognize that generated data may not fully capture the complexities of real-world clinical scenarios. However, obtaining data that accurately reflects real-world clinical interactions, such as human radiologists revising draft reports, is exceptionally challenging. This requires extensive recording of clinical procedures, which raises issues of copyright and patient privacy. Managing such records, even if available, as a published dataset poses significant logistical and ethical challenges.
> 2. **LLMs demonstrate strong medical knowledge**: LLMs have shown impressive capabilities in medical tasks. For example, as mentioned in our paper, ChatGPT has been tested and successfully passed the US Medical Licensing Exams (USMLE). Additionally, a recent study [1] found that ChatGPT outperformed human doctors in diagnosing illnesses, achieving a 90% accuracy rate. These results highlight that LLMs possess sufficient knowledge for medical applications.
> 3. **Human validation of generated data distribution**: We conducted detailed human validation of the generated data. As noted in line 262 of our paper, over 95% of the generated data were deemed plausible by medical professionals, ensuring a high degree of reliability.
> 4. While synthetic data serves as a practical alternative, we acknowledge in our limitations section that collecting real-world data is always preferable whenever feasible. This remains an important area for future exploration.
>
> [1] Goh E, Gallo R, Hom J, et al. Large Language Model Influence on Diagnostic Reasoning: A Randomized Clinical Trial. JAMA Netw Open. 2024;7(10):e2440969. doi:10.1001/jamanetworkopen.2024.40969
>
>
> # Applying to Other Domains
> > Furthermore, the paper focuses on radiology report generation with a proposed method specifically designed and optimized for this purpose. It remains unclear how the pipeline used to produce this new dataset and the new method, DeMMo, can be applied to other domains, which would be of interest to a broader ICLR audience.
>
> Thank you for the comment. Our primary focus in this paper is on radiology report generation, and the proposed method, DeMMo, is specifically designed and optimized for this task. While we recognize the potential applicability of our approach to other domains, exploring these applications is currently beyond the scope of this work. However, we believe the methodology we present can be adapted to other domains, and we plan to explore these possibilities in future research.
>
>
> # Correction
> > What is "correction" in Line 264?
>
> In our paper "report correction" and "report revision" are used interchangeably. We have updated this in our paper
>
> # Template-based Writing
> > Is templated-based writing actually useful? Do radiologists write reports following templates? Figure 1 also shows a relatively low acceptance rate.
>
> Our medical professionals have reviewed and validated the tasks and corresponding data, affirming their clinical relevance and usefulness. The relatively low acceptance rate may stem from the limitations of ChatGPT in handling structured text data, which increases the likelihood of errors such as content misplacement or omissions, as stated in our error analysis in Section 4.3.

---

> ### Author Response · Authors · 2024-11-27
>
> # Comparing different tasks
> > Table 2 seems to demonstrate that using previous radiology reports and medical records is not helpful (compared to no context). Any insights?
>
> While Table 2 suggests that training with previous reports and medical records does help our model outperform baselines on most metrics in the no-context task, we still want to emphasize that our main aim is to provide a unified framework for report generation tasks with contextual input. Each task is designed to be relatively independent and not directly comparable to others. Although our model benefits from contextual information, our focus is on creating a comprehensive benchmark rather than optimizing performance across tasks by training on data from other tasks.

---

> > ### Comment · Reviewer_tNne · 2024-11-27
> >
> > Thanks to the authors for the response! After reviewing the response, the concern listed under "Weaknesses" still stands. Therefore, I will leave my assessment unchanged.

---

### Official Review · Reviewer_zCYn · 2024-11-04

**Soundness:** 4
**Presentation:** 2
**Contribution:** 2
**Rating:** 6
**Confidence:** 4

**Summary:**

The paper identifies a problem with current radiology report generation datasets: they fail to evaluate models’ performance in a real-world workflow, where models might be asked to incorporate different contextual information and generate a wide range of text formats instead of a simple, image-to-text, radiology report. Its answer to this problem is a novel benchmark, MIMIC-R3G, derived from MIMIC-CXR, which evaluates generative models on five tasks: no-context report generation, generation using medical records as context, using previous visits as context, report revision, and templated generation. The paper further introduces a baseline for this benchmark: a model that fuses medical visual representations with Flamingo, and shows its competitive performance on the benchmark compared to some popular radiology report generation models.

**Strengths:**

I agree with the paper’s central claim that current modeling and evaluation approaches do not fully reflect models’ potential use cases in a real workflow. Evaluating models’ ability to use contextual information is a relatively new but important concern for radiology report generation.

The additional tasks to MIMIC-CXR are well-motivated, and the methodology to generate them using GPT-4 is rigorous. Another strength of this method is that it is scalable because the data for the new tasks are automatically generated using GPT-4. Most importantly, the paper validates this procedure using annotations from radiologists.

I enjoyed reading section 5! The notation is very clear and the diagram added to my understanding of the method. Furthermore, the results show the baseline’s effectiveness, as it achieves top performance in most tasks in the benchmark.

**Weaknesses:**

I think the most significant weakness of this paper is the low impact of the contribution. While evaluation that reflects a real-world workflow is an important problem, I think that the benchmark offers a limited approach. MIMIC-CXR already contains information about previous visits. The medical records, although validated as plausible, are still synthetic data. It is unclear how systems trained on synthetic data will perform in real settings. The report revision and template generation tasks still function within the single-turn interaction setting, whereas real-world interactions will likely be multi-turn.

Aside from that, I noticed several typos, such as:
* Line 191: I think $I_i$ is supposed to be $R_i$.
* Line 202: “posses”
* Line 430: “difference tasks”

There are more typos than just these. Please address them accordingly!

**Questions:**

Questions:
* Line 343: The queries are concatenated with the visual features. This means later on the queries will just attend to themselves. What benefit does this offer?

Some more suggestions (these are just suggestions to improve the paper and they do not factor into my evaluation of the paper):
* Line 222: I think “Quality Control of the Generation” should be a separate subsection because it is about validating the dataset.

---

> ### Author Response · Authors · 2024-11-27
>
> Thank you for the constructive review! Please see our response below:
>
> # Contribution
> > I think the most significant weakness of this paper is the low impact of the contribution. While evaluation that reflects a real-world workflow is an important problem, I think that the benchmark offers a limited approach. MIMIC-CXR already contains information about previous visits. The medical records, although validated as plausible, are still synthetic data. It is unclear how systems trained on synthetic data will perform in real settings. The report revision and template generation tasks still function within the single-turn interaction setting, whereas real-world interactions will likely be multi-turn.
>
> 1. **Previous visit context**: We acknowledge that using previous visits as context is not novel and can be derived from MIMIC-CXR metadata. However, its inclusion ensures a comprehensive benchmark by covering an important aspect of real-world workflows.
>
> 3. **Multi-turn interactions**: Our primary goal is to provide a unified format for report generation tasks with contextual inputs. While multi-turn interaction is a valuable direction, generating and formatting such data is non-trivial and resource-intensive. However, our proposed data and tasks can be seamlessly adapted for simple multi-turn interactions. For instance, users could provide multiple revision instructions in sequential steps for step-by-step report refinement, or use medical records to generate a free-text report followed by a template for the next round of generation. This flexibility highlights the adaptability of our approach to multi-round applications.
>
> [1] Goh E, Gallo R, Hom J, et al. Large Language Model Influence on Diagnostic Reasoning: A Randomized Clinical Trial. JAMA Netw Open. 2024;7(10):e2440969. doi:10.1001/jamanetworkopen.2024.40969
>
>
>
> # Typos
> > Aside from that, I noticed several typos
>
> Thanks for pointing these out, we have updated our paper accordingly.
>
>
> # Concatenation with Queries
> > Line 343: The queries are concatenated with the visual features. This means later on the queries will just attend to themselves. What benefit does this offer?
>
> Following the original architecture of Flamingo, the queries and visual features are concatenated to form the keys and values for attention. As explained in the Flamingo paper, this approach was based on empirical results, where they "found it to perform slightly better."

---

### Official Review · Reviewer_1v6h · 2024-11-04

**Soundness:** 2
**Presentation:** 2
**Contribution:** 2
**Rating:** 5
**Confidence:** 4

**Summary:**

The paper presents MIMIC-R3G, a new benchmark dataset designed to advance automatic radiology report generation by integrating real-world clinical contexts and instructions. Radiology report generation traditionally approached as a captioning task, often lacks adaptability to complex clinical workflows that involve physician-specific instructions and patient history. The authors address this limitation by proposing a structured pipeline that utilizes large language models (LLMs) to produce high-quality instructional data, integrating contextual elements like prior patient records and lab results. MIMIC-R3G extends MIMIC-CXR by introducing five tasks that emulate practical report generation scenarios, such as report revision and template-based generation. The authors also introduce a baseline model, DeMMo (Domain-enhanced Multimodal Model), tailored for medical contexts, which incorporates a specialized medical vision encoder and pathological guidance, enhancing report accuracy by leveraging detailed domain knowledge. Extensive experiments indicate that incorporating context and instruction substantially improves report generation performance, setting a new standard for real-world radiology applications.

**Strengths:**

1. Automatic data generation pipeline leveraging GPT-4 to produce high-quality instructions and context for report generation tasks.
2. Extensive evaluation of state-of-the-art methods using the proposed benchmark datasets.
3.  Introduction of the Domain-enhanced Multimodal Model (DeMMo), a novel architectural modification suited for radiology report generation.

**Weaknesses:**

1. The synthetic data generation process, while innovative, may not fully capture the nuanced and complex nature of real-world clinical interactions and instructions.
2. The paper lacks a direct comparison of the generated data and model outputs with real-world clinical data.
3. Motivation behind architectural changes in DeMMo remains unclear.
4. Experiments are shown only on the proposed dataset, which does not validate the efficacy of the proposed model DeMMo.

**Questions:**

1. Could you please clarify the paragraph of report revision, there seems to be a typo where report is denoted with C.
2. How did you ensure that the types and frequencies of errors introduced in the report revision task accurately reflect real-world errors made by human radiologists or existing AI systems?
3. There are other available radiology report datasets. Does DeMMo trained on MIMIC-R3G showcase better performance on other datasets, i.e., does pertaining on MIMIC-R3G enhance performance on MIMIC-CXR, and other available datasets like IU-XRay, ROCO.
4. Did you keep all the modules learnable while training DeMMo on MIMIC-R3G dataset? If yes, please mention the training details, i.e., whether the LLM was trained or not?

**Details Of Ethics Concerns:**

No ethics review needed as per my understanding.

---

> ### Author Response · Authors · 2024-11-27
>
> Thank you so much for the constructive review! Please see our response below:
>
> # Synthetic Data vs Real-world Data
> > The synthetic data generation process, while innovative, may not fully capture the nuanced and complex nature of real-world clinical interactions and instructions.
>
> 1. **Collecting real-world clinical distributions is extremely challenging**: We recognize that generated data may not fully capture the complexities of real-world clinical scenarios. However, obtaining data that accurately reflects real-world clinical interactions, such as human radiologists revising draft reports, is exceptionally challenging. This requires extensive recording of clinical procedures, which raises issues of copyright and patient privacy. Managing such records, even if available, as a published dataset poses significant logistical and ethical challenges.
> 2. **LLMs demonstrate strong medical knowledge**: LLMs have shown impressive capabilities in medical tasks. For example, as mentioned in our paper, ChatGPT has been tested and successfully passed the US Medical Licensing Exams (USMLE). Additionally, a recent study [1] found that ChatGPT outperformed human doctors in diagnosing illnesses, achieving a 90% accuracy rate. These results highlight that LLMs possess sufficient knowledge for medical applications.
> 3. **Human validation of generated data distribution**: We conducted detailed human validation of the generated data. As noted in line 262 of our paper, over 95% of the generated data were deemed plausible by medical professionals, ensuring a high degree of reliability.
> 4. While synthetic data serves as a practical alternative, we acknowledge in our limitations section that collecting real-world data is always preferable whenever feasible. This remains an important area for future exploration.
>
> [1] Goh E, Gallo R, Hom J, et al. Large Language Model Influence on Diagnostic Reasoning: A Randomized Clinical Trial. JAMA Netw Open. 2024;7(10):e2440969. doi:10.1001/jamanetworkopen.2024.40969
>
>
> # Comparison with Real-world Data
> > The paper lacks a direct comparison of the generated data and model outputs with real-world clinical data.
>
> We have not encountered publicly available real-world clinical datasets that align with the specific settings of our proposed tasks, such as generating reports with templates as context or modifying existing information in a report. If the reviewer is aware of any published clinical datasets with similar task formatting, we would greatly appreciate the suggestion and will gladly include a comparative analysis in our work.
> Moreover, to ensure the validity of our approach, we have taken steps to validate the generated data through human evaluation, as detailed in our paper. Over 95% of the generated data was deemed plausible by medical professionals, providing confidence in its quality. We believe this evaluation partially addresses the gap where direct comparisons with real-world data are not feasible.
> We further compute the percentage of positive samples for each label using the annotations provided in MIMIC-CXR, and the statistics indicate that all the percentages differ by no more than 3%, showing that the disease distribution of generated data aligns with real-world data.
>
>
> | Label                      | MIMIC-CXR % | 600-sample % |
> | -------------------------- | ----------- | ------------ |
> | No Findings                | 17.4        | 15.4         |
> | Enlarged Cardiomediastinum | 4.6         | 3.9          |
> | Cardiomegaly               | 26.5        | 26.9         |
> | Lung Lesion                | 3.8         | 5.2          |
> | Lung Opacity               | 31.8        | 31.1         |
> | Edema                      | 20.9        | 23.9         |
> | Consolidation              | 6.7         | 7.5          |
> | Pneumonia                  | 10.2        | 10.2         |
> | Atelectasis                | 22.4        | 23.9         |
> | Pneumothorax               | 3.4         | 3.6          |
> | Pleural Effusion           | 32.9        | 31.5         |
> | Pleural Other              | 1.9         | 1.3          |
> | Fracture                   | 2.7         | 1.6          |
> | Support Devices            | 34.4        | 33.4         |

---

> ### Author Response · Authors · 2024-11-27
>
> # DeMMo Motivation
> > Motivation behind architectural changes in DeMMo remains unclear.
>
> It is important to note that our main contribution lies in the introduction of new tasks and the dataset, while DeMMo serves as a baseline method for benchmarking this dataset. Given the page constraints, we believe we have provided adequate descriptions and a clear model figure that should help readers understand the model’s functionality and design rationale.
>
> Additional details are provided in Appendix Section B, to support the general understanding of the baseline method. The design is also supported by comprehensive ablation studies in Appendix section C.4. We encourage the reviewer to check the appendix for further details on the model design and ablation of each component. Please let us know if there are specific aspects of the model that remain unclear and require further clarification.
>
>
> # Experiment Dataset
> > Experiments are shown only on the proposed dataset, which does not validate the efficacy of the proposed model DeMMo.
>
> Again, we have not encountered other publicly available datasets with similar task formats or settings. If the reviewer could suggest any such datasets, we would be glad to include additional experiments and comparisons to further validate the efficacy of our proposed baseline model.
>
> # Report Revision Typo
> > Could you please clarify the paragraph of report revision, there seems to be a typo where report is denoted with C.
>
> If the reviewer is referring to L185-186, this is not a typo. $C_i$ is the report that the user wants to modify, which is a part of the context input. In report revision task, models should take report $C_i$ and instruction $I_i$ as context and generate $R'_i=R_i$, which is the ground truth report from original report generation dataset.
>
>
> # Reflecting Real-world Errors
> > How did you ensure that the types and frequencies of errors introduced in the report revision task accurately reflect real-world errors made by human radiologists or existing AI systems?
>
> Again as stated in response titled **Synthetic Data vs Real-world Data**, collecting real-world distribution is extremely challenging. Moreover, there should be low impact on the distribution shift due to relatively simple task of revision. This task focuses more on a much simpler task of revision based on human instructions, and should be independent to the distribution of how AI systems or human radiologists would make errors in the diagnosis. Therefore, we believe that our current pipeline, which includes human validation, is the most practical and valid approach. Also as stated in our paper, for the report revision task, 194 out of 200 samples (97%) were considered highly plausible by medical professionals. With that being said, we have also discussed this as one of our limitations in appendix.
>
>
> # Other Datasets
> > There are other available radiology report datasets. Does DeMMo trained on MIMIC-R3G showcase better performance on other datasets, i.e., does pertaining on MIMIC-R3G enhance performance on MIMIC-CXR, and other available datasets like IU-XRay, ROCO.
>
> Thank you for the suggestion. Our primary focus is on MIMIC-CXR, given its status as the largest and most comprehensive dataset for radiology report generation, and our aim is not to improve performance on different tasks or different datasets. While we have not conducted experiments on other datasets like IU-XRay or ROCO, we anticipate that generating context data for those datasets using our method would similarly enhance performance on their corresponding tasks, given the emphasis on task-specific context understanding and report generation.
>
>
> # Training Details
> > Did you keep all the modules learnable while training DeMMo on MIMIC-R3G dataset? If yes, please mention the training details, i.e., whether the LLM was trained or not?
>
> As shown in Figure 2, for our domain-enhanced perceiver resampler module we only tune the projection layer, the adaptation prompt, and the tanh gate for medical visual features. Additionally, we keep the gated cross-attention layers of Flamingo learnable, and keep the original LLaMa layers frozen. We have updated the description in section 5.

---

### Official Review · Reviewer_Rak3 · 2024-11-04

**Soundness:** 2
**Presentation:** 3
**Contribution:** 2
**Rating:** 3
**Confidence:** 4

**Summary:**

This paper addresses the limitations of existing medical report generation models in capturing important contextual information beyond radiology images by proposing a real-world radiology report generation model (R3G) and a new benchmark dataset, MIMIC-R3G. MIMIC-R3G comprises five sub-tasks that reflect various clinical needs, each designed to generate reports based on medical images using specific instructions and contextual information as inputs.

The five sub-tasks are as follows:

1. No Context Report Generation: Generates a report based on medical images and basic instructions as inputs.
2. Report Revision: Produces a revised report using modification instructions and the current report as inputs.
3. Template-based Report Generation: Generates a structured report by filling in a template based on provided instructions and a blank template as inputs.
4. Previous Visit as Context: Generates a report based on instructions and prior visit records to reflect the current report with historical context.
5. Medical Records and Lab Tests as Context: Generates a comprehensive report that incorporates patient medical records and lab test results provided as contextual input.

The dataset was constructed through a data generation pipeline that creates instructions and contextual information specific to each sub-task, allowing for the training and evaluation of the R3G model with the MIMIC-R3G dataset.

In conclusion, this paper introduces the MIMIC-R3G dataset, an automated data generation pipeline, and DeMMo, a Flamingo-based model, as a baseline for the R3G benchmark dataset.

**Strengths:**

A key strength of this study is its attempt to provide a new benchmark dataset specifically based on chest x-rays. Addressing the following weaknesses would further enhance its value and impact.

**Weaknesses:**

[About benchmark dataset]

The proposed benchmark evaluates five distinct tasks using consistent metrics and baselines, which obscures the unique objectives and distinctions of each task. Task-specific metrics and baselines are needed to highlight these differences better.

For the Report Generation task, the benchmark dataset CXR-PRO already exists to address temporal-changing hallucinations.

The Previous Visit as Context task lacks novelty, as it aligns closely with the inherent characteristics of chest x-ray reports. The MS-CXR-T dataset, which already enables focused comparisons of temporal changes in diseases, limits the distinctiveness of this task.

Furthermore, in the 3) Report Revision and 4) Template tasks, the primary focus is on compliance with instructions despite the use of chest X-ray images. For a more robust assessment, these tasks should include detailed and consistent evaluations reflecting clinical standards. Including an LLM model baseline and focusing on instruction adherence rather than traditional NLG or CE metrics would enhance the evaluation.

Finally, 5) for the Medical Records and Lab Tests as Context task, the impact of incorporating real medical records on diagnostic content should be emphasized. For example, evaluations could assess if diagnoses change from negative to positive when medical records are included.

[About experiment part]

The study's main contribution is the introduction of a new benchmark dataset. While DeMMo, based on Flamingo, is an incremental improvement, additional experiments are needed for thorough validation. Current comparisons are primarily with generalist models (e.g., ChatCAD+, Med-Flamingo) trained on diverse medical images, making DeMMo's advantage over these models unsurprising. However, comparisons with chest x-ray specialist models are limited to CvT2DistillGPT2. Broader comparisons with other chest x-ray-focused models, including instruction-finetuned models (ex. LLM-CXR: Instruction-Finetuned LLM for CXR Image Understanding and Generation), are essential to establish DeMMo’s validity as a baseline.

These refinements would strengthen the benchmark's clarity and alignment with clinical objectives.

**Questions:**

N/A

---

> ### Author Response · Authors · 2024-11-27
>
> Thank you so much for the constructive review! Please see our response below:
>
> # Temporal issues in report generation
> > For the Report Generation task, the benchmark dataset CXR-PRO already exists to address temporal-changing hallucinations.
>
> Our work aims to include previous report as a context for report generation, so that current generated report can more accurately refer to the previous report regarding any changes. CXR-PRO addresses temporal changing hallucinations by removing references to previous reports to make each report generation independent, which is orthogonal to our work.
>
>
> # Previous visit as context
> > The Previous Visit as Context task lacks novelty, as it aligns closely with the inherent characteristics of chest x-ray reports. The MS-CXR-T dataset, which already enables focused comparisons of temporal changes in diseases, limits the distinctiveness of this task.
>
> While we acknowledge that including the previous visit as context is not a novel task and that corresponding data can be easily retrieved using MIMIC-CXR metadata, we included it for the sake of comprehensiveness in the benchmark. Our goal is to encourage generated reports to accurately reference the previous report when describing changes, which is distinct from the focus of the MS-CXR-T dataset. Unlike MS-CXR-T, which emphasizes image classification and sentence similarity tasks, our task prioritizes the generation of coherent and temporally aware radiology reports, making it complementary rather than overlapping.
>
>
> # Instruction adherence and task-specific metrics
> > The proposed benchmark evaluates five distinct tasks using consistent metrics and baselines, which obscures the unique objectives and distinctions of each task. Task-specific metrics and baselines are needed to highlight these differences better.
> > Furthermore, in the 3) Report Revision and 4) Template tasks, the primary focus is on compliance with instructions despite the use of chest X-ray images. For a more robust assessment, these tasks should include detailed and consistent evaluations reflecting clinical standards. Including an LLM model baseline and focusing on instruction adherence rather than traditional NLG or CE metrics would enhance the evaluation.
>
> Our goal is to design benchmarks that encourage models to adhere to specific instructions in clinical scenarios while grounding their outputs in relevant chest X-ray images. We believe that CE and NLG metrics remain critical sufficient, as they directly evaluate the clinical correctness of generated reports and whether context input is properly considered for generation. For example, in the Report Revision task, if the model fails to generate an accurately revised report, this will be reflected in the CE metric. Conversely, if the model produces a clinically correct report from the images but disregards the provided context, this will be evident through discrepancies in NLG metrics. The same reasoning applies to Template task as well. We have included various LLM baselines such as GPT-4V and ChatCAD in our experiments.
> That said, we acknowledge that additional metrics and baselines, such as focusing on instruction-specific adherence, could further enhance the evaluation for individual tasks. However, our primary objective in this work is to establish a unified framework and set of metrics for report generation with contextual inputs. Expanding the evaluation to include more specialized metrics is an excellent direction for future work, and we will consider this in subsequent iterations.

---

> ### Author Response · Authors · 2024-11-27
>
> # Impact of incorporating real medical records
> > Finally, 5) for the Medical Records and Lab Tests as Context task, the impact of incorporating real medical records on diagnostic content should be emphasized. For example, evaluations could assess if diagnoses change from negative to positive when medical records are included.
>
> Thank you for the suggestion. While incorporating medical records is intended to serve as supplementary information, with the report primarily based on image interpretation, we have observed instances where the inclusion of medical records influenced the diagnosis. For example, in one case, the diagnosis of cardiomegaly changed from negative to positive, likely due to the presence of a documented history of heart disease in the medical records. Below is an inference example illustrating this effect:
>
> Medical Records:
> > 1. chronic obstructive pulmonary disease (copd) or other interstitial lung diseases
> > 2. heart disease
> > 3. coronary artery disease
> > 4. severe degenerative joint disease, possibly osteoarthritis, in the shoulders.
>
> Inference with no context:
> > pa and lateral views of the chest provided. left chest wall pacer device is again noted with leads extending to the region the right atrium and right ventricle. midline sternotomy wires and mediastinal clips are again noted. there is a linear density projecting over the right lower lung which is unchanged and may represent scarring. no convincing evidence for pneumonia or edema. no large effusion or pneumothorax. **the heart is top-normal in size**. mediastinal contour is normal. no acute bony abnormalities.
>
> Inference with context:
> > the patient is status post coronary artery bypass graft surgery. a dual-lead pacemaker/icd device appears unchanged, again with leads terminating in the right atrium and ventricle, respectively. **the heart is moderately enlarged**. the mediastinal and hilar contours appear unchanged. there is similar blunting of each costophrenic sulcus suggesting scarring or trace pleural effusions. right lateral pleural thickening appears unchanged. otherwise, there is marked improvement in a right basilar opacity since the prior examination. there is no pneumothorax.
>
> Ground truth:
> > since the prior exam, there appears to be increased interstitial prominence, although no overt pulmonary edema. stable bronchiectasis and scarring is again noted at the right base. there is no dense consolidation. there is no pleural effusion or pneumothorax. severe cardiomegaly is present. a pacemaker is in place with wires in unchanged position. the patient is status post a cabg. the sternal wires are intact. there are severe degenerative changes of the bilateral shoulders.
>
>
> # Contribution
> > The study's main contribution is the introduction of a new benchmark dataset. While DeMMo, based on Flamingo, is an incremental improvement, additional experiments are needed for thorough validation. Current comparisons are primarily with generalist models (e.g., ChatCAD+, Med-Flamingo) trained on diverse medical images, making DeMMo's advantage over these models unsurprising. However, comparisons with chest x-ray specialist models are limited to CvT2DistillGPT2. Broader comparisons with other chest x-ray-focused models, including instruction-finetuned models (ex. LLM-CXR: Instruction-Finetuned LLM for CXR Image Understanding and Generation), are essential to establish DeMMo’s validity as a baseline.
>
> It is important to note that our main contribution lies in the introduction of new tasks and the dataset, while DeMMo serves as a baseline method for benchmarking this dataset. We agree that more baseline comparisons would enhance the validity of our proposed method, and update the results of LLM-CXR in our paper.

---

> > ### Comment · Reviewer_Rak3 · 2024-11-28
> >
> > I have reviewed the authors' responses and gained a better understanding through the examples and explanations they provided, but the weaknesses I highlighted still stand, so I will not be adjusting my scores.

---

### Meta-Review · Area_Chair_JxYD · 2024-12-21

**Metareview:**

The paper introduces MIMIC-R3G, a benchmark dataset and data generation pipeline, enabling radiology report generation with instructions and context, alongside a newly proposed multimodal model for improved clinical performance.

Strengths
- The paper introduces a well-motivated benchmark (MIMIC-R3G) with five clinically relevant tasks, and the proposed pipeline leverages large language models to generate high-quality synthetic data validated by radiologists
- The Domain-enhanced Multimodal Model (DeMMo) demonstrates improved performance over baselines

Weaknesses
- The use of synthetic data (generated via LLMs) raises concerns about its alignment with real-world clinical scenarios
- The proposed dataset and model are only evaluated within the MIMIC-R3G framework, with no comparisons to other available datasets which significantly undermines the claim of generalizability
- The proposed model DeMMo lacks task-specific adaptations, reducing the focus on optimizing performance for individual subtasks

**Additional Comments On Reviewer Discussion:**

During the rebuttal, authors acknowledged limitations and concerns raised by the reviewers (e.g. reliance of synthetic data only, use of unified metrics across different tasks, and lack of comparison with CXR-specific models), but only promised to resolve these issues in the future work.

---

### Decision · Program_Chairs · 2025-01-22

Reject